# Autoinhibited kinesin-1 adopts a hierarchical folding pattern

Zhenyu Tan[1,2], Yang Yue[3], Felipe Leprevost[4], Sarah Haynes[4], Venkatesha Basrur[4], Alexey I Nesvizhskii[4,5], Kristen J Verhey[3], Michael A Cianfrocco[2,6]*

[1]Department of Biophysics, University of Michigan, Ann Arbor, United States; [2]Life Sciences Institute, University of Michigan, Ann Arbor, United States; [3]Department of Cell & Developmental Biology, University of Michigan, Ann Arbor, United States; [4]Department of Pathology, University of Michigan, Ann Arbor, United States; [5]Department of Computational Medicine and Bioinformatics, University of Michigan, Ann Arbor, United States; [6]Department of Biological Chemistry, University of Michigan, Ann Arbor, United States

**\*For correspondence:**
mcianfro@umich.edu

**Abstract** Conventional kinesin-1 is the primary anterograde motor in cells for transporting cellular cargo. While there is a consensus that the C-terminal tail of kinesin-1 inhibits motility, the molecular architecture of a full-length autoinhibited kinesin-1 remains unknown. Here, we combine crosslinking mass spectrometry (XL-MS), electron microscopy (EM), and AlphaFold structure prediction to determine the architecture of the full-length autoinhibited kinesin-1 homodimer (kinesin-1 heavy chain [KHC]) and kinesin-1 heterotetramer (KHC bound to kinesin light chain 1 [KLC1]). Our integrative analysis shows that kinesin-1 forms a compact, bent conformation through a break in coiled-coil 3. Moreover, our XL-MS analysis demonstrates that kinesin light chains stabilize the folded inhibited state rather than inducing a new structural state. Using our structural model, we show that disruption of multiple interactions between the motor, stalk, and tail domains is required to activate the full-length kinesin-1. Our work offers a conceptual framework for understanding how cargo adaptors and microtubule-associated proteins relieve autoinhibition to promote activation.

## eLife assessment

This paper will be of significant interest to the research community working on cytoplasmic transport and microtubule motors, offering **important** insights into the structural arrangement of autoinhibited Kinesin-1. The paper reports a structural model of full-length kinesin-1 describing its autoinhibitory mechanism using cryo-EM, Alphafold structural predictions, cross-linking, and mass spectrometry. The data are of high quality and together offer a **compelling** model for how Kinesin-1 is autoinhibited, indicating that auto-inhibition does not rely on the IAK motif alone but on a more extensive intramolecular interface.

## Introduction

The spatiotemporal regulation of organelle positioning is critical for proper cellular function. The movement of organelles and other intracellular cargo is largely driven by the cytoskeletal motor proteins (*Hirokawa et al., 2009*; *Reck-Peterson et al., 2018*). Among them, conventional kinesin-1 is the major anterograde motor in cells that is essential for transporting mitochondria (*Schwarz, 2013*), ER, and Golgi-derived vesicles (*Hirokawa et al., 2009*) to microtubule plus ends, typically located at the cell periphery.

Kinesin-1 can exist in homodimeric and heterotetrameric forms. Kinesin-1 in the dimeric form is composed of two identical kinesin heavy chains (KHC) that can be divided into three parts: motor, stalk, and tail domains (*Figure 1A*; *Hirokawa et al., 1989*). The N-terminal motor domain functions as an engine that converts the chemical energy from ATP to the mechanical force to power kinesin walking along microtubules (*Jon Kull et al., 1996*). Downstream of the motor domain, a series of coiled-coils dimerize to form the stalk. The regulatory tail at the C-terminus is largely disordered (*Seeger et al., 2012*). In the heterotetramer form, two copies of the kinesin light chain (KLC) associate with the KHC stalk through its predicted N-terminal coiled-coil domain (*Diefenbach et al., 1998*;

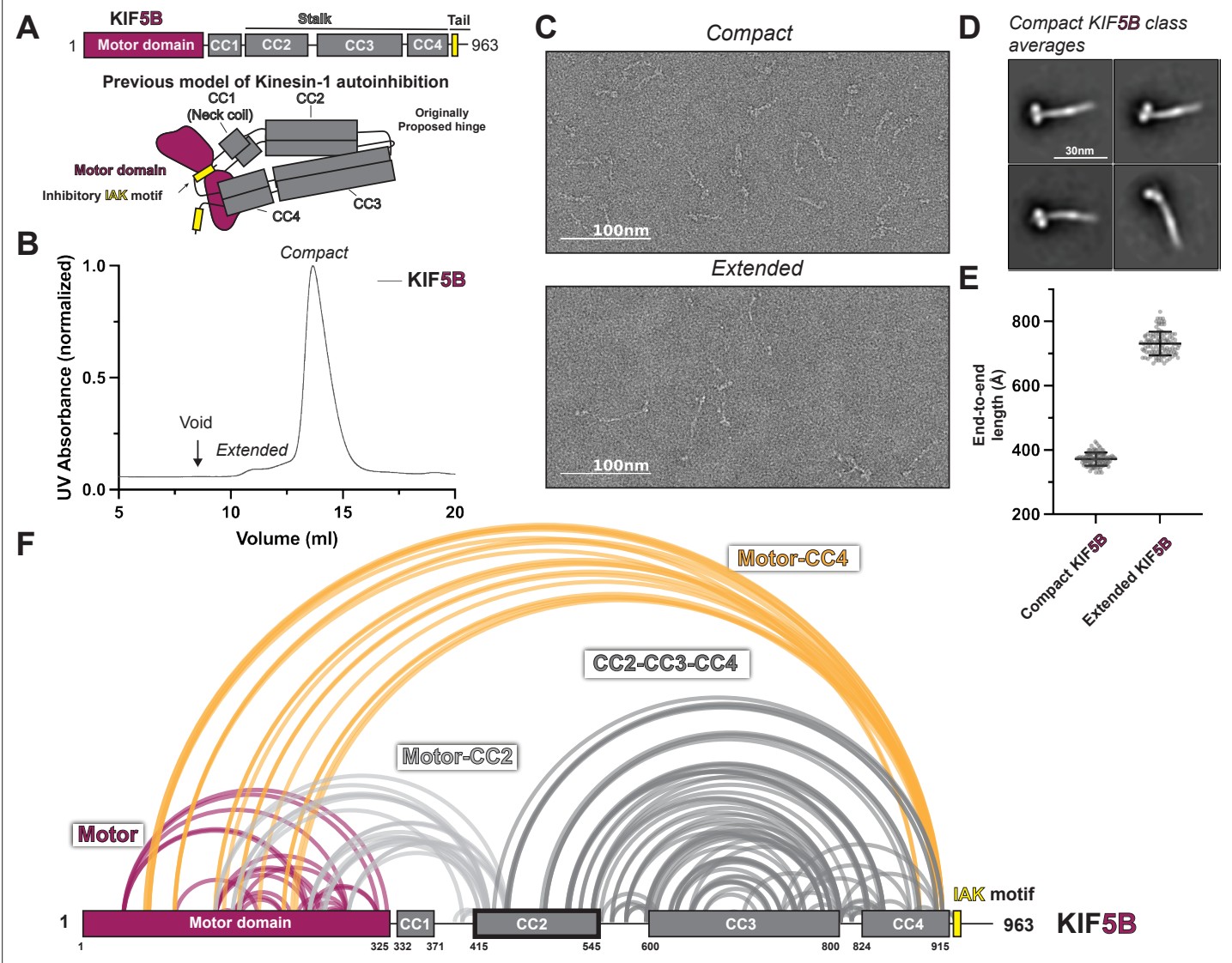

**Figure 1.** KIF5B adopts a hierarchical folding pattern. (**A**) Overview of the KIF5B domain diagram and the previous model of kinesin-1 autoinhibition. (**B**) Size-exclusion chromatography profile of KIF5B. (**C**) Negative staining electron microscopy (EM) analysis of KIF5B in two conformations. (**D**) Class averages of KIF5B in compact state. (**E**) End-to-end measurements of KIF5B in two states (N = 100). (**F**) Crosslinked lysine pairs in KIF5B were mapped onto the domain diagram and divided into four groups.

The online version of this article includes the following source data and figure supplement(s) for figure 1:

**Source data 1.** XL-MS data for KIF5B.

**Source data 2.** XL-MS data for KIF5C.

**Figure supplement 1.** KIF5C adopts a hierarchical folding pattern.

**Figure supplement 2.** Structural analysis and purification of KIF5B and KIF5C.

**Figure supplement 3.** Negative stain electron microscopy (EM) of KIF5B and KIF5C.

*Verhey et al., 1998*). The C-terminus of KLC consists of the TPR domain, which functions as a cargo binding platform to link kinesin-1 to various cellular cargo (*Cockburn et al., 2018*; *Pernigo et al., 2013*).

Human kinesin-1 contains various isoforms, and dysfunction of them causes diseases. In humans, KHC is encoded by three genes: KIF5A, KIF5B, and KIF5C, which share a very similar motor domain and stalk with minor differences in the C-terminal tail, whereas KLC has four isoforms (KLC1–4). These isoforms showed distinct tissue expression patterns. KIF5B and KLC2 are ubiquitously expressed, whereas KIF5A, KIF5C, and KLC1 are neuron-enriched (*Kanai et al., 2000*; *Rahman et al., 1998*). Mutations in KIF5A and KIF5C result in defects in axonal transport and thus can cause neurodevelopmental and neurodegenerative pathologies (*Crimella et al., 2012*; *Nicolas et al., 2018*; *Poirier et al., 2013*; *Willemsen et al., 2014*).

When not bound to cargo, kinesin-1 exists in an autoinhibited state. Several early studies have proposed a model where the motility of KHC is regulated through the direct interaction between the motor and tail domains (*Coy et al., 1999*; *Dietrich et al., 2008*; *Friedman and Vale, 1999*; *Stock et al., 1999*). This interaction is facilitated by the folding of KHC at a break between CC2 and CC3, called 'hinge,' which positions an isoleucine–alanine–lysine (IAK) motif in the tail domain to bind in between the motor domains (*Figure 1A*; *Kaan et al., 2011*). The association between tail and motor domains restricts the relative movement of the motor domains, inhibiting KHC's enzymatic and microtubule-binding activities (*Hackney and Stock, 2000*; *Kaan et al., 2011*).

Despite this prevailing model, recent data suggest that mutating the IAK motif or deleting the hinge does not increase the processivity of kinesin-1, making the current model controversial. Mutations in the IAK motif have been reported to disrupt kinesin-1 autoinhibition in cells, enzymatic assays, and single-molecule assays performed with cell lysates (*Cai et al., 2007a*; *Kelliher et al., 2018*; *Seiler et al., 2000*). However, recent in vitro single-molecule measurement using purified protein suggests that mutations in IAK did not result in a substantial increase in processive motility but increased the landing rate (*Chiba et al., 2022*). Similarly, deletion of the hinge was traditionally demonstrated to increase the ATPase activity and frequency of processive KHC dimer motility (*Coy et al., 1999*; *Friedman and Vale, 1999*; *Kelliher et al., 2018*), while recent data suggest that KHC without the hinge tends to oligomerize and shows little increase in processivity (*Chiba et al., 2022*). The contradictory data regarding the IAK motif and hinge indicate that kinesin-1 autoinhibition involves more than these two elements.

Kinesin-1 motility is further regulated by KLCs, where the incorporation of KLCs into heterotetrameric kinesin-1 motors inhibits microtubule binding in vitro and in vivo (*Verhey et al., 1998*). For instance, cellular FRET experiments implied that KLC keeps kinesin-1 in the folded state but separates the motor domain further apart (*Cai et al., 2007a*). Recently, in vitro single-molecule assays further demonstrated that the presence of KLCs suppresses the kinesin-1 motility (*Chiba et al., 2022*). However, it remains unclear whether KLCs stabilize the inhibited state of the heavy chain or promote the formation of a new structural state. Furthermore, how cargo binding to KLCs activates kinesin-1 is still an open question.

Microtubule-associated proteins function as an additional layer to regulate motor protein activity. For instance, MAP7 increases the landing rate and processivity of kinesin-1 by binding to coiled-coil 2 and recruiting kinesin-1 onto microtubules (*Ferro et al., 2022*; *Hooikaas et al., 2019*; *Monroy et al., 2018*). MAP7 contains an N-terminal microtubule-binding domain and a C-terminal kinesin-binding domain connected by an unstructured linker (*Hooikaas et al., 2019*). Interestingly, the MAP7 kinesin binding domain alone is sufficient to promote the landing rate and processivity of kinesin-1 (*Hooikaas et al., 2019*), suggesting that binding MAP7 to CC2 disrupts the autoinhibited state. Furthermore, truncated kinesin-1 homodimer KIF5B(1–560) that lacks the hinge and downstream elements can be further activated by MAP7 kinesin binding domain (*Hooikaas et al., 2019*), suggesting that autoinhibition is still present in KIF5B(1–560), which the current model can not explain.

Recently, a study combining AlphaFold structure prediction and negative staining EM proposed a new hinge position to allow the stalk to fold on itself (*Weijman et al., 2022*). While this study is an important prediction of KHC bends, there is minimal data to provide an overall description of how kinesin-1 adopts a folded, inhibited state. Moreover, this study did not address how the folded conformation relates to motor domain inhibition.

Here, we combined crosslinking mass spectrometry (XL-MS), negative stain electron microscopy (EM), and AlphaFold structure prediction to probe the conformation of autoinhibited full-length kinesin-1 homodimer and kinesin-1 heterotetramer. First, we performed structural studies with negative stain EM to characterize the folded conformation of kinesin-1 homodimers and heterotetramers. Both biochemical and structural analysis showed that kinesin-1 exists in folded and extended conformations. To determine the folding pattern of kinesin-1, we combined protein structure prediction using AlphaFold and XL-MS to derive an integrative molecular model of the autoinhibited full-length kinesin-1. For the kinesin-1 homodimer, we found that it does not simply fold in half via the originally proposed hinge but rather adopts a hierarchical folding pattern. A newly identified hinge in coiled-coil 3 allows the stalk to fold back on itself. The motor domain also scaffolds back to dock onto coiled-coil 2, positioning it in a head-to-tail conformation. XL-MS analysis of the heterotetramer revealed that the KLC does not change the folding pattern of the heavy chain, indicating that the light chain does not promote a new structural state but stabilizes the inhibited state. We validated our proposed autoinhibition model via mutagenesis studies to show that disruption of multiple regions is required to fully activate the full-length kinesin-1. We demonstrate that autoinhibition requires more interactions than the inhibitory tail peptide, providing the first comprehensive explanation of autoinhibition. Our model offers a conceptual framework for understanding how activation factors such as MAP7 and cargo adaptors relieve autoinhibition to promote full activation.

## Results

### Autoinhibited kinesin-1 adopts a hierarchical folding pattern

To provide a detailed view of the interactions within autoinhibited kinesin-1, we set out to directly probe the architecture of autoinhibited kinesin-1 via negative stain EM and XL-MS. First, we selected human isoforms KIF5B and KIF5C as our targets which have ubiquitous and neuronal-specific expression, respectively. KIF5B and KIF5C have four coiled-coil (CC) domains, CC1 to CC4 (*Figure 1A*, *Figure 1—figure supplements 1A and 2A*), where CC1 corresponds to the neck coil, and CC2-CC4 form the stalk. Following CC4 is an unstructured tail domain that contains the inhibitory IAK motif. The region between CC2 and CC3 has a low coiled-coil prediction probability and was thus originally proposed to be the hinge where kinesin-1 folds in half.

We purified full-length recombinant KIF5B and KIF5C for negative stain EM structural studies (*Figure 1—figure supplement 2B*). As seen previously (*Chiba et al., 2022*; *Weijman et al., 2022*), following size-exclusion chromatography, KIF5B and KIF5C showed the presence of two populations with short and long retention times (*Figure 1B*, *Figure 1—figure supplement 1B*). Interestingly, the proportion of compact versus extended form for KIF5B and KIF5C differs. KIF5C seems to have a larger proportion of extended form compared to KIF5B, which could be due to the sequence difference between these two isoforms. After collecting fractions from both peaks and crosslinking the proteins using bis(sulfosuccinimidyl)suberate , we imaged each peak fraction using negative staining EM (*Figure 1C and E, Figure 1—figure supplements 1C and E and 3*). Samples from the first peak adopt a fully extended conformation with a length of around 80 nm, consistent with the previously reported extended kinesin length from other species (*Amos, 1987*; *Hirokawa et al., 1989*; *Kuznetsov et al., 1988*). In contrast, samples from the second peak have a length corresponding to half of the fully extended kinesin, indicative of the folded autoinhibited state. Further analysis of the folded kinesin showed 2D class averages containing a thick rod with two globular domains, which we attribute to the folded stalk and motor domains (*Figure 1D*, *Figure 1—figure supplement 1D*).

To obtain higher-resolution structural information, we performed XL-MS on both KIF5B and KIF5C. We collected crosslinked samples from the second peak, digested them with trypsin, and analyzed them using tandem mass spectroscopy (MS/MS). We identified crosslinked peptides using pLink software (*Chen et al., 2019*) and presented all high-confidence crosslinks (E value < 0.01) as lines between crosslinked lysine pairs in the KIF5B and KIF5C domain diagram (*Figure 1F*, *Figure 1—figure supplement 1F*).

Our XL-MS data show that KIF5B and KIF5C exhibit similar folding patterns (*Figure 1F*, *Figure 1—figure supplement 1F*). Unexpectedly, we discovered that kinesin-1 engages in four intramolecular interactions that suggest a hierarchical folding pattern: (1) crosslinks within the motor domain (*Figure 1F*, *Figure 1—figure supplement 1F*, purple lines), (2) motor-to-tail crosslinks that surprisingly

involve residues at the C-terminal end of CC4 (K903 and K909 in KIF5B or K905 and K911 in KIF5C) but not the IAK region itself (*Figure 1F*, *Figure 1—figure supplement 1F*, yellow lines), (3) motor-to-CC2 crosslinks (*Figure 1F*, *Figure 1—figure supplement 1F*, light gray lines), and (4) crosslinks within CC2, CC3, and CC4. Together, these crosslinks show that kinesin-1 does not fold back via the originally proposed hinge between CC2 and CC3, but rather, the folding of the stalk domain is mediated by a break within the CC3, which folds the C-terminus of CC3 and CC4 onto CC2 and the N-terminus of CC3 (*Figure 1F*, *Figure 1—figure supplement 1F*, dark gray lines).

## Integrative modeling reveals the molecular architecture of autoinhibited kinesin-1

To generate a model for the autoinhibited kinesin-1 homodimer, we used AlphaFold2 predictions (*Jumper et al., 2021*) of kinesin-1 fragments and integrated them using information from XL-MS and EM. Given that KIF5B and KIF5C have very similar folding patterns (*Figure 1*, *Figure 1—figure supplement 1*), we selected KIF5B for the following analysis.

To understand how the stalk folds on itself, we used AlphaFold-Multimer (*Evans et al., 2022*) to predict a KIF5B homodimer fragment containing CC2, CC3, CC4, and tail (residues 401–963). The homodimeric stalk model from AlphaFold (*Figure 2—figure supplement 1A*) is comparable to the coiled-coil prediction results (*Figure 1—figure supplement 2A*) but with two main exceptions. First, AlphaFold suggests that the originally proposed hinge between CC2 and CC3 is not a gap between coiled-coils but rather is a tetrameric coiled-coil linked by a short loop. Second, AlphaFold suggests that there is a breaking point within CC3. We surmise that the originally proposed hinge between CC2 and CC3 arose from coiled-coil prediction artifacts, and the real hinge that enables KHC folding is the breaking point in CC3, consistent with recent work (*Weijman et al., 2022*). Therefore, we divided CC3 into CC3a and CC3b and labeled the breaking point connecting CC3a and CC3b as the hinge (*Figure 2A*). This coiled-coil domain arrangement can also be seen in the predicted KIF5C stalk structure (*Figure 2—figure supplement 1B*).

We used these predicted models and a negative stain 3D reconstruction of KIF5B to generate a model for the inhibited full-length KIF5B (*Figure 2A*). We used AlphaFold-Multimer (*Evans et al., 2022*) to independently predict the structure of four KIF5B fragments that are chosen based on the locations of coiled-coil domain breaks: motor-CC1-CC2 (aa 1–540), CC2-CC3a (aa 401–690), CC3b (aa 691–820), and CC4-tail (aa 821–963). For the motor-CC1-CC2 (aa 1–540) part, we found that the AlphaFold prediction results in a motor domain conformation similar to the tail-bound structure (PDB:2Y65; *Kaan et al., 2011*; *Figure 2—figure supplement 2B*). Interestingly, the prediction shows that the motor domain scaffolds back to dock one of the motor domains along the CC2 (*Figure 2—figure supplement 2A*), resulting in an asymmetric conformation. We then used the CC2 region to superimpose the CC2-CC3a fragment with the motor-CC1-CC2 fragment. Finally, with guidance from our negative stain 3D map, we placed the CC3b (aa 691–820) and CC4-tail (aa 821–963) regions along CC2 and CC3a and inserted the end of CC4 between the motor domains. The resulting assembled model of full-length KIF5B agrees with the low-resolution map from negative staining EM (*Figure 2B*) except the CC1 domain. We noticed that the CC1 domain extended beyond the contour of the 3D volume, which could indicate that our predicted CC1 structure did not reflect the true CC1 conformation.

We then used the XL-MS data to validate and extend this model of autoinhibited KIF5B. As a control, we confirmed that crosslinked residues within the motor domain identified in the XL-MS data analysis pipeline are compatible with the known structure of the motor domain (*Figure 2—figure supplement 2C*). Indeed, 22 of 25 crosslinked pair distances are within the theoretical limit of Bfigure supplement 3 crosslinker distance constraint (24 Å) (*Merkley et al., 2014*; *Figure 2—figure supplement 2D*), indicating the fidelity of our pipeline.

To verify the folded architecture of the stalk, we mapped the XL-MS data onto the model of autoinhibited KIF5B dimer and found that the crosslinked residues confirm the interactions between CC2 and CC4 and between CC3a and CC3b (*Figure 2C*, panel 1). Thus, XL-MS confirmed the hinge position and the organization of the stalk in the folded autoinhibited kinesin-1.

Folding of the stalk domain positions the end of the CC4 adjacent to the motor domains. Our XL-MS data shows that CC4 has multiple contacts with the motor domain relevant to ATP and microtubule binding (*Figure 2C*, panel 2). We observed that lysine 903 and 909 (K903, K909) made multiple

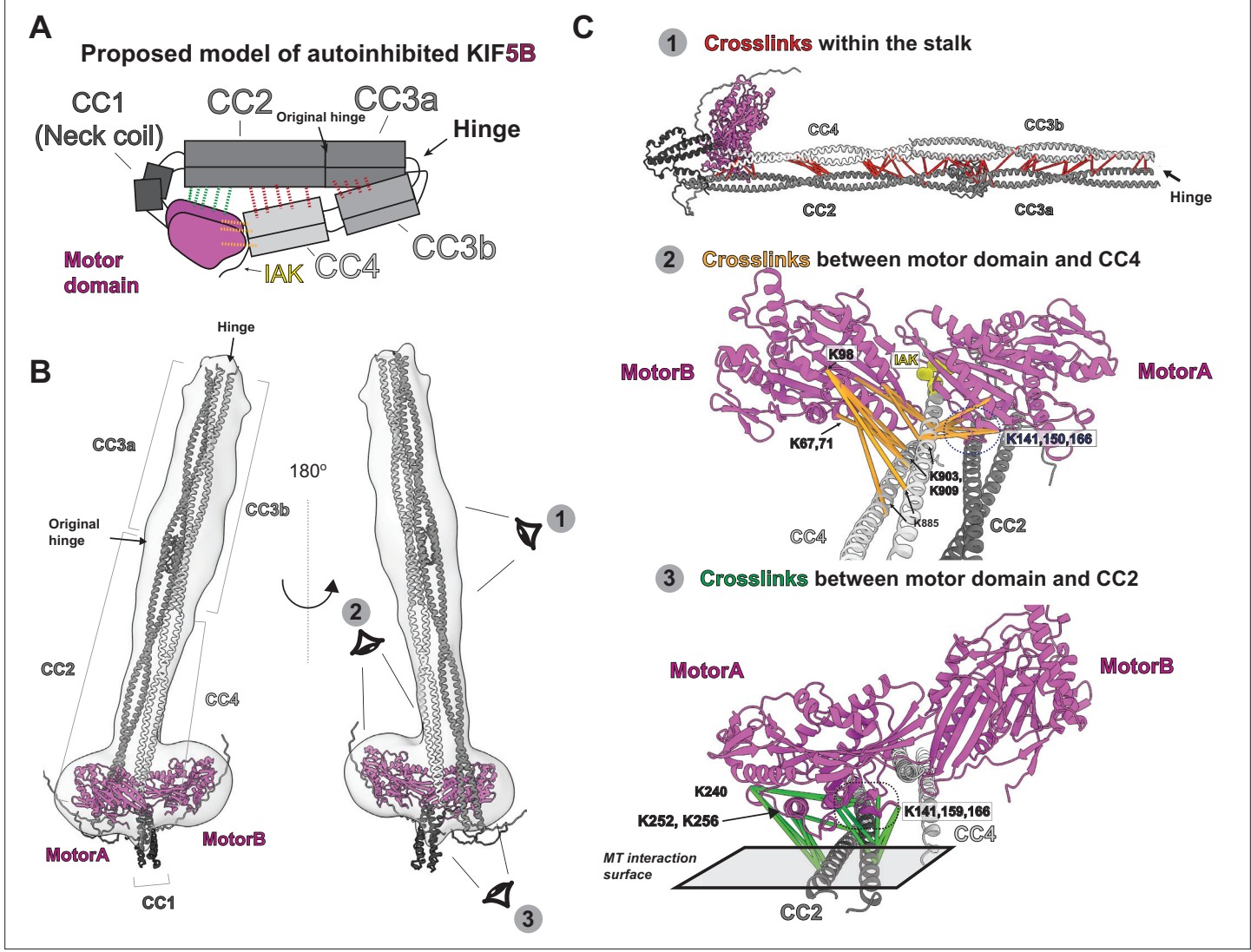

**Figure 2.** Integrative modeling reveals the molecular architecture of autoinhibited KIF5B. (**A**) The proposed model of autoinhibited KIF5B in cartoon diagram. The dashed lines indicates the crosslinked pairs. (**B**) A model of autoinhibited KIF5B via the integrative modeling. The gray density is the 3D reconstruction from the negative staining electron microscopy (EM). Three eye icons showed the viewing directions in (**C**). (**C**) Three groups of crosslinked pairs were mapped onto the KIF5B model.

The online version of this article includes the following figure supplement(s) for figure 2:

**Figure supplement 1.** The AlphaFold predicted structures of KIF5B and KIF5C stalk and tail.

**Figure supplement 2.** Predicted KIF5B motor domain resembles the tail peptide-bound state.

**Figure supplement 3.** The stalk crosslinks to multiple points on the motor domain.

contact points to the motor domain within the distance range of 10–40 Å. Considering that our BS3 crosslinker has a maximum crosslink distance of 24 Å, we assigned the crosslinked residues from CC4 to two different motor domains, motorA and motorB. We see that K903 and K909 crosslink to motor domain residues K67 and K71 from motorB α1 helix and K98 from motorB loop5 (*Figure 2C*, panel 2, *Figure 2—figure supplement 3B*). These two elements are involved in kinesin ATPase cycles (*Atherton et al., 2014*), indicating that CC4 and tail may regulate the enzymatic activity of the motor domain. Furthermore, K903 and K909 engage with motorA β5 and loop8 (K141, K150, and K166) (*Figure 2C*, panel 2, *Figure 2—figure supplement 3A*), which span the binding interface between kinesin and β-tubulin (*Atherton et al., 2014*), indicating that the autoinhibited kinesin is incapable of landing on microtubules. Finally, K903 and K909 were crosslinked to K213, K222, and K226 in motorB β6 and β7. Interestingly, we did not see tail residues IAK crosslinked to the motor domain as previously

seen in structural studies (*Kaan et al., 2011*), which could be due to the lack of neighboring groups that can crosslink with the lysine in the motif. Considering that the tail is close to the motor, we believe the IAK peptide should lie at the interface of motorA and motorB. However, further work is needed to define the location of IAK in the inhibited kinesin-1.

Finally, we used the XL-MS data to verify the asymmetric arrangement of the motor domains along CC2. Our 3D modeling suggests that motorA is adjacent to both CC2 and CC4 while motorB is only close to CC4 (*Figure 2B and C*, *Figure 2—figure supplement 2A*). We, therefore, mapped the cross-linked pairs between only motorA and CC2 to see whether this asymmetric arrangement agrees with the XL-MS data, resulting in a distance range of 10–35 Å. We found that the lysine residues in CC2 (K420, K427, K430, K448, and K450) make extensive contact with two clusters in motorA (*Figure 2C*, panel 3, *Figure 2—figure supplement 3A*): β5-loop8 and α4-loop11, both of which are involved in microtubule binding. β5-loop8 (residues K141, K159, and K166) forms the β-tubulin binding surface, whereas α4-loop11 (residues K240, K252, and K256) directly binds α-tubulin. These results suggest that, like CC4, CC2 precludes the motor domain from binding to microtubules, although CC2 blocks microtubule binding through two different contact points.

Taken together, our integrative modeling approach revealed the molecular architecture of autoinhibited kinesin-1. Our model verified the new hinge position in between CC3a and CC3b and showed that the motor domain scaffolded back to dock on to CC2. The CC4 thus can insert in between the motor domain. We found that CC2 and CC4 sterically block the microtubule-binding interface, suggesting that autoinhibited kiesin-1 is incompatible with microtubule binding. Our model agrees with most of the crosslinks but the exact conformation of the motor domain and tail needs further study.

## Kinesin light chains form extensive contacts with the heavy chain while maintaining the autoinhibited heavy chain folding pattern

To determine the role of KLCs in promoting autoinhibition, we utilized the EM and XL-MS data to characterize the architecture of the kinesin-1 heterotetramer. We recombinantly expressed and purified heterotetramers of KIF5B-KLC1 and KIF5C-KLC1 (*Figure 3—figure supplement 2A and B*) and found that KIF5C-KLC1 showed two distinct populations on the chromatogram, similar to previous reports (*Weijman et al., 2022*; *Figure 3B*). The first peak is the fully extended conformation with a length of around 75 nm (*Figure 3C and E*), whereas the second peak is the compact form with a length of around 35 nm. 2D class averages from the compact population revealed a structure comparable to kinesin-1 homodimers with a pair of globular densities corresponding to the motor domains at one end of the class averages (*Figure 3D*). Unlike kinesin-1 homodimers, however, we saw a new C-shaped density along the stalk in various orientations from 80% class averages, which we believe are the light chains (*Figure 3D*). Unlike KIF5C-KLC1, KIF5B-KLC1 was purified mainly in the compact form from the size-exclusion chromatography (*Figure 3—figure supplement 1B*) with a length of 35 nm (*Figure 3—figure supplement 1C and E*). The class averages revealed a similar overall architecture compared to the KIF5C-KLC1 (*Figure 3—figure supplement 1D*), with the motor domain and stalk easily identified along with the C-shaped density on the stalk.

To probe the structure of autoinhibited kinesin-1 tetramer, we took the fractions corresponding to the compact form and crosslinked them using bis(sulfosuccinimidyl)suberate (BS3) (*Figure 3—figure supplement 2A and B*). The crosslinked samples were analyzed using the same pipeline as the kinesin-1 dimer. All high-confidence crosslinks (E value < 0.01) were plotted on the kinesin-1 heterotetramer domain diagram (*Figure 3F*, *Figure 3—figure supplement 1F*).

Our XL-MS data show that the KHC folding pattern was not changed with the addition of the light chains (*Figure 3F*, *Figure 3—figure supplement 1F*). This suggests that the KLC did not induce a new structural state of the heavy chain. We observed that KLC1 made more extensive contacts with heavy chain isoform KIF5C than KIF5B (*Figure 3F*, *Figure 3—figure supplement 1F*). The N-terminus of KLC1 makes contact between the CC3 and CC4 in KIF5B-KLC1 and KIF5C-KLC1, which is consistent with the previously reported binding interface (*Verhey et al., 1998*). The TPR domain of KLC1 made extensive contacts with the motor domain, CC2, and CC4 in KIF5C-KLC1 (*Figure 3F*), while it was only sparsely linked to the motor domain and CC4 in the KIF5B-KLC1 (*Figure 3—figure supplement 1F*). We also observed intra-KLC1 crosslinks (*Figure 3—figure supplement 2C and D*) from both samples, though with slightly different patterns, indicating a difference in light chain conformations.

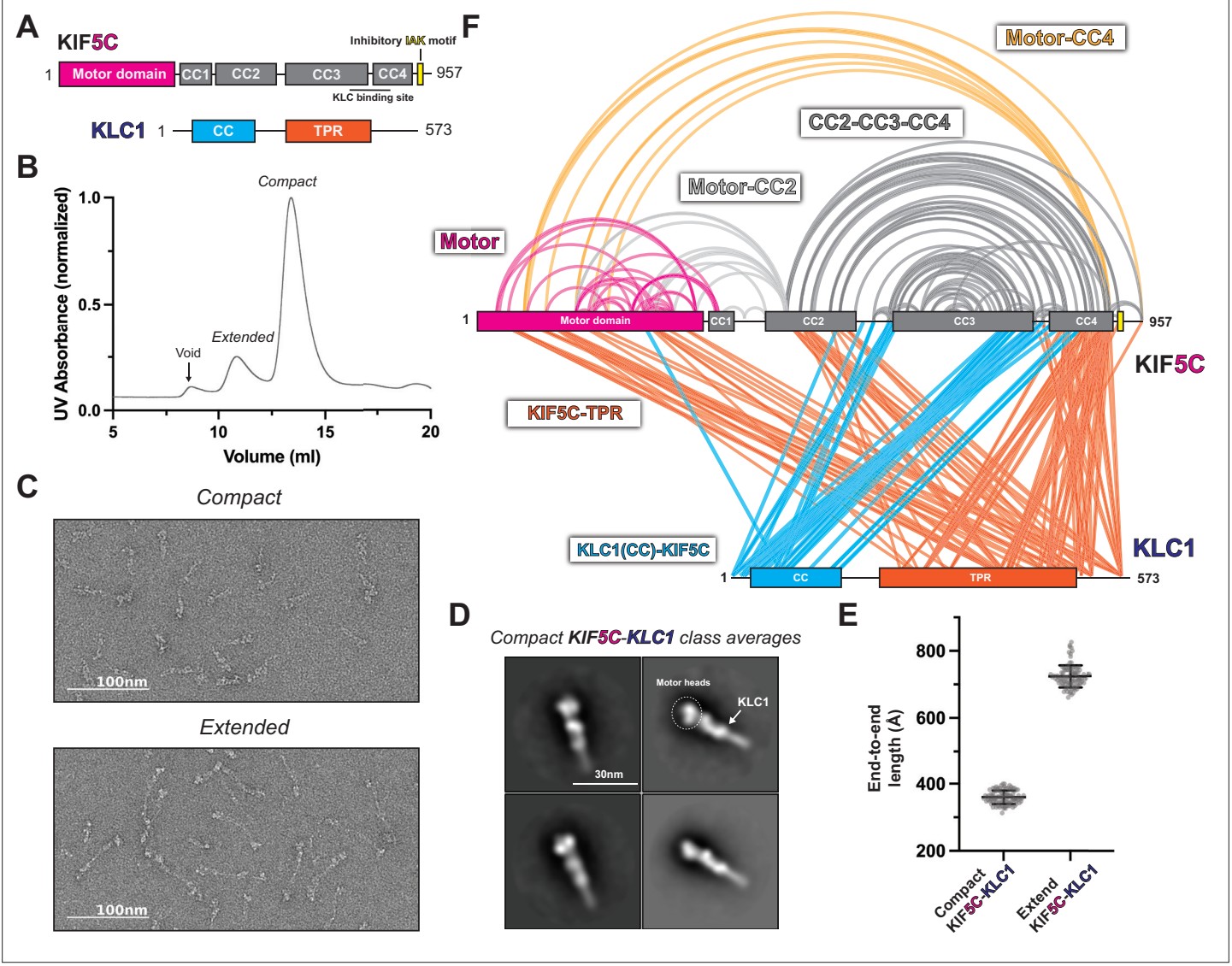

**Figure 3.** KLC1 makes extensive contacts with KIF5C within the inhibited state. (**A**) Overview of KIF5C-KLC1 domain diagram. (**B**) Size-exclusion chromatography profile of KIF5C-KLC1. (**C**) Negative staining electron microscopy (EM) analysis of Kif5c-KLC1 in two states. (**D**) Class averages of KIF5C-KLC1 in compact conformation. (**E**) End-to-end length measurements of KIF5C-KLC1 in two states (N = 100). (**F**) Crosslinked lysine pairs in KIF5C-KLC1 were mapped onto the domain diagram.

The online version of this article includes the following source data and figure supplement(s) for figure 3:

**Source data 1.** XL-MS data for KIF5C-KLC1.

**Source data 2.** XL-MS data for KIF5B-KLC1.

**Figure supplement 1.** KLC1 maintains the folding pattern of KIF5B.

**Figure supplement 2.** SDS-PAGE analysis of kinesin-1 heterotetramer (KIF5B-KLC1 and KIF5C-KLC1) and the intra-KLC1 crosslinks.

Specifically, we found crosslinks between the C-terminus of the KLC1 coiled-coil and the TPR domain, as well as intra-TPR crosslinks. The exact conformation of KLC1 in the heterotetramer needs further investigation.

## Kinesin light chains stabilize the folded conformation of the kinesin heavy chain

We set out to model a structure of the autoinhibited kinesin-1 heterotetramer (*Figure 4A*) with the input from low-resolution EM reconstruction, XL-MS, and predicted structure fragments from

AlphaFold. While we noticed that the folding patterns of KIF5B and KIF5C were not changed with the addition of KLC1, KLC1 made more contacts with KIF5C (*Figure 3F*, *Figure 3—figure supplement 1F*), which could be due to the different light chain conformation in the two tetramers. In our 3D modeling, we focused on KIF5C-KLC1 given the extensive crosslinks observed in this sample.

To generate our models, we began by using AlphaFold Multimer (*Evans et al., 2022*) to predict a structure of the KIF5C:KLC1 interface consisting of CC2 to the tail of KIF5C (aa 401–957) and the N-terminal domain of KLC1 (aa 1–200) containing the coiled-coil (*Figure 4—figure supplement 1B*). The predicted structure showed a similar coiled-coil domain arrangement within KIF5C as the predicted KIF5C homodimer structure (*Figure 2—figure supplement 1B*). Interestingly, the KIF5C-KLC1 interface was predicted to be an eight-helix bundle, also seen in a previous report (*Weijman et al., 2022*). The eight-helix bundle consists of a dimer of four coiled-coils formed by three short N-terminal helices from the KLC1 and the C-terminus of CC3b from KIF5C. The same KLC1 binding configuration can also be seen in the predicted structure of KIF5B-KLC1 (*Figure 4—figure supplement 1A*).

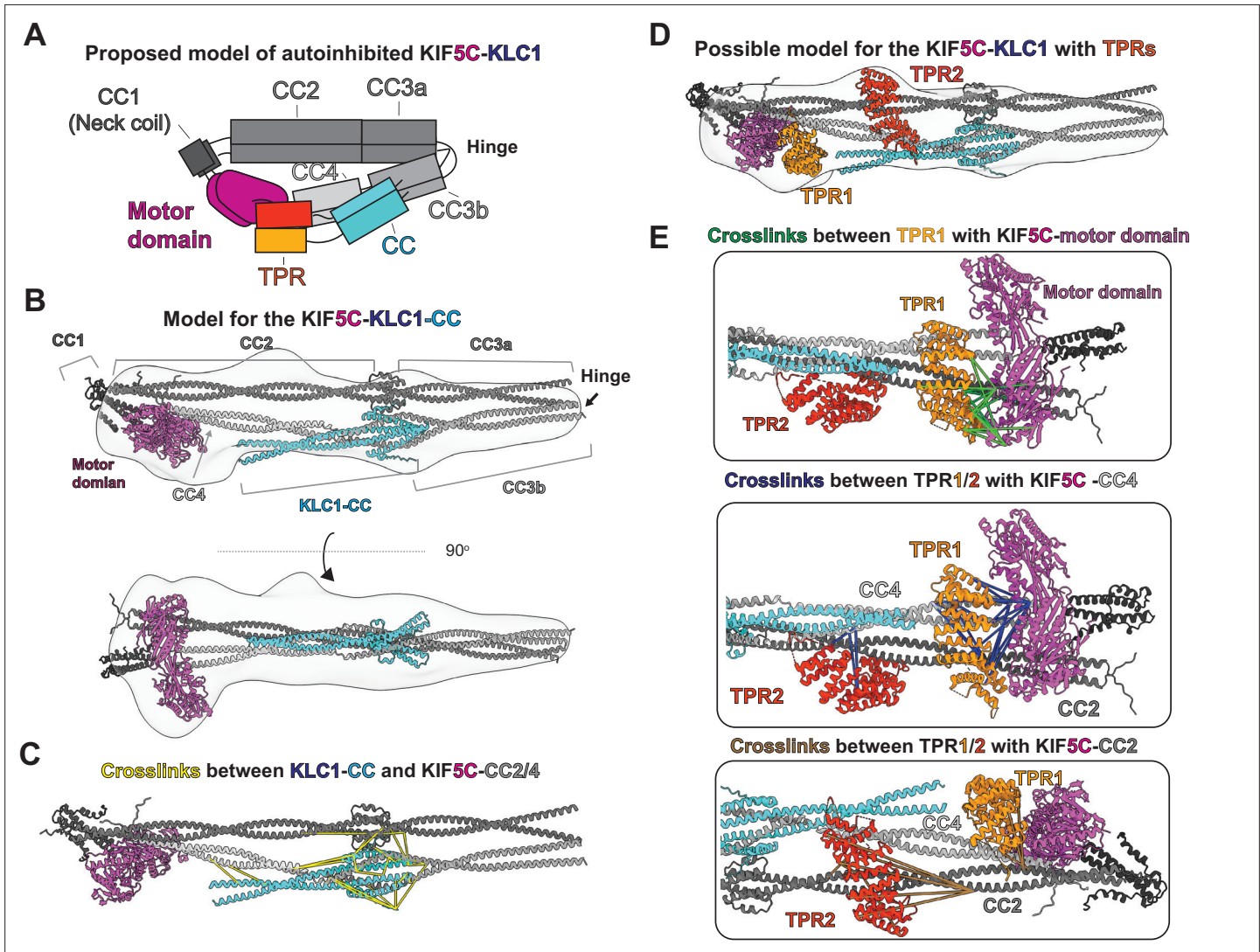

**Figure 4.** Integrative modeling reveals molecular architecture of autoinhibited KIF5C-KLC1. (**A**) The proposed model of autoinhibited KIF5C-KLC1 in cartoon diagram. (**B**) A model of KIF5C-KLC1(CC) via integrative modeling. The gray density is the low-resolution map from negative staining electron microscopy (EM). (**C**) The crosslinked pairs between KLC1-CC and KIF5C stalk, shown in the structure model. (**D**) The locations for the TPR domains in the KIF5C-KLC1. (**E**) Crosslinked lysine pairs between TPR domains and KIF5C.

The online version of this article includes the following figure supplement(s) for figure 4:

**Figure supplement 1.** The AlphaFold predicted structure of KIF5B/C stalk and KLC1.

Building off of our model of the KIF5C-KLC1 interface, we developed a model for the KIF5C-KLC1 without the TPR domain as a starting point. We separately predicted the structure of CC3b-CC4-tail-KLC1(CC), CC2-CC3a, and motor-CC1-CC2 via AlphaFold Multimer and manually combined them with the guidance from a negative stain 3D reconstruction and XL-MS. We modeled the configuration of the KIF5C same as the KIF5B dimer, as they have almost identical crosslinking patterns. The resulting KIF5C-KLC1(CC) model agrees with the low-resolution reconstruction (*Figure 4B*).

To validate the KIF5C-KLC1 interface in our model, we mapped the crosslinked residues between KLC1(CC) and KIF5C and found that they were distributed within the eight-helix bundle with a distance range from 5 to 50 Å (*Figure 4C*). The longest crosslinks came from the first methionine in KLC1, which exists in an unstructured and flexible region. The N-terminus of KLC1 also makes contact with the tetrameric coiled-coil knot between the CC2 and CC3a. Our XL-MS data support the predicted binding interface configuration, but the exact structure needs further exploration.

Finally, to map the location of the TPR domains within KIF5C-KLC1, we used the model of KIF5C-KLC1(CC) as a starting point. After docking the KIF5C-KLC1(CC) into the EM map, we noticed two unoccupied extra densities. One is adjacent to the CC2 and CC4, the other is on top of the motor domain. We, therefore, placed the two structures of the TPR domain (PDB:3NF1) (*Zhu et al., 2012*) separately to the two unoccupied densities, with TPR1 on top of the motor domain and TPR2 adjacent to the CC2 and CC4 (*Figure 4D*).

We used the crosslinks between KIF5C and the TPR domain to verify these placements. The TPR domain crosslinked to three regions in KIF5C, which are the motor domain, CC2, and CC4 (*Figure 3F*). To best satisfy the distance constraints, we assigned crosslinks between the motor domain and TPR to TPR1 and noticed that TPR1 touches the ATP binding pocket (*Figure 4E*, green lines). The crosslinks between the C-terminus of CC4 (K877, K884, K905, and K911) and TPR were mainly assigned to TPR1 as they are in close proximity. The remaining crosslinks between the N-terminus of CC4 and TPR were mapped onto TPR2, consistent with the location of TPR2 along CC2 and CC4 (*Figure 4E*, blue lines). At last, the crosslinks between TPR and CC2 were mapped onto TPR2 (*Figure 4E*, brown lines). We noticed that several long-distance crosslinks could not be explained by our current model, especially the crosslinks between CC2 and TPR2 (*Figure 4E*, brown lines). Taken together, our XL-MS results suggest that our placement of two TPR domains likely reflects the real location but the exact conformation of TPR needs to be determined.

## Disrupting the hierarchical folding of kinesin-1 activates motility

Our integrative modeling suggests that kinesin-1 autoinhibition involves the hierarchical folding of the protein into a compact, inhibited state. To test this model, we generated mutations along the kinesin-1 heavy chain and tested whether there was a resulting increase in motility. Using KIF5B, we designed a series of mutations and deletions and performed single-molecule motility assays using an established cell lysate assay (*Cai et al., 2007b*), where the motility of the fluorescently KIF5B proteins along microtubules was visualized utilizing total internal reflection fluorescence (TIRF) microscopy. (*Figure 5A*, *Figure 5—figure supplement 3*).

To set a baseline for fully activated kinesin-1, we measured the motile properties of truncated KIF5B constructs tagged at their C-terminus with monomeric NeonGreen. Full-length KIF5B showed very few motility events (*Figure 5B*), confirming that the kinesin-1 dimer is autoinhibited. In contrast, a minimal dimer construct consisting of only the motor and CC1 (neck coil) domains without any regulatory elements to lock it into inhibited sate, KIF5B(1–420), displays a significantly increased landing rate and processive events on microtubules (*Figure 5B*), although its dwell time on microtubules is relatively short (*Figure 5—figure supplement 1B*). A longer construct that includes CC2, KIF5B(1–565), and is widely used in the field to investigate kinesin-1 motility, has a lower landing rate and longer dwell time compared to KIF5B(1–420) but is still dramatically activated compared to full-length KIF5B (*Figure 5B*, *Figure 5—figure supplement 1A and B*). The lower landing rate relative to KIF5B(1–420) suggests that some degree of autoinhibition may still be present in KIF5B(1–565). Consistent with this possibility, XL-MS demonstrates that crosslinks between the motor domain and CC2 remain for this construct (*Figure 5—figure supplement 1D*), suggesting that its motor domain is partially prevented from binding microtubules. The motor-CC2 crosslinks in KIF5B(1–565) also reveal that the folding of the motor domain onto CC2 and folding of the remaining stalk are independent.

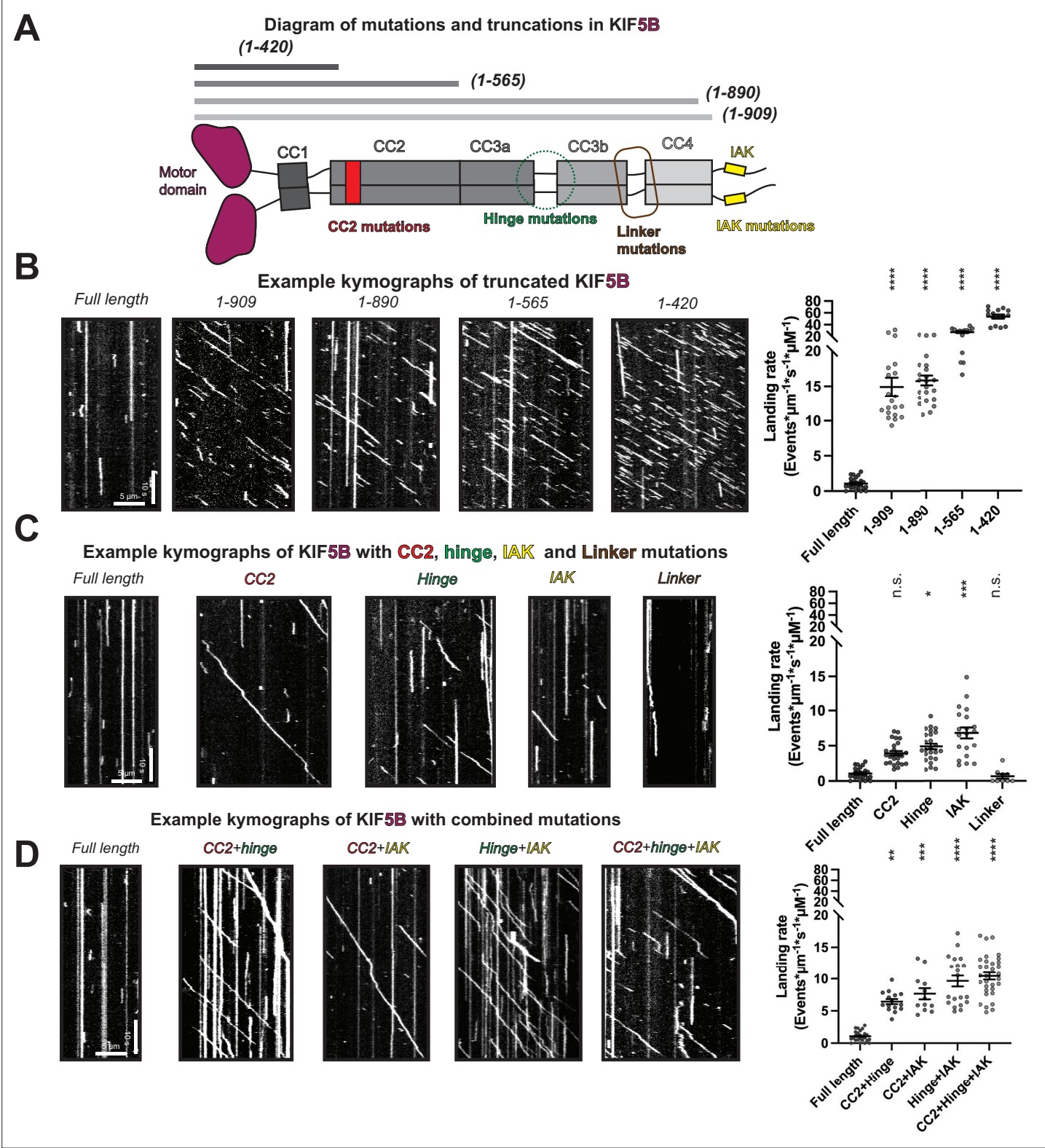

**Figure 5.** Disruption of the hierarchical folding activates KIF5B motility in vitro. (**A**) The designed KIF5B mutations and truncations shown on the domain diagram. (**B**) Example kymographs of KIF5B with different truncations. And the corresponding landing rate. Landing rate (events/μm/s/μM): Lines show mean ± SEM: 1.07 ± 0.16 (WT), 14.92 ± 1.30 (1–909), 15.75 ± 0.70 (1–890), 26.87 ± 1.65 (1–565), 53.42 ± 3.47 (1–420). N = 2; n = 25, 20, 23, 15 and 14 MTs. One-way ANOVA followed by Dunnett's test. ****p<0.0001. (**C**) Example kymographs of KIF5B with single mutation and the corresponding landing rate. Landing rate (events/μm/s/μM): Lines show mean ± SEM: 1.07 ± 0.16 (WT), 3.93 ± 0.32 (CC2), 4.93 ± 0.42 (hinge), 6.90 ± 0.78 (IAK), 0.67 ± 0.32 (linker).

*Figure 5 continued on next page*

*Figure 5 continued*

N = 2; n = 25, 25, 25, 20, and 9 MTs. One-way ANOVA followed by Dunnett's test. *p<0.05, ***p<0.001, n.s., not significant. (**D**) Example kymographs of KIF5B with combined mutations and the corresponding landing rate. Landing rate (events/μm/s/μM): Lines show mean ± SEM: 1.07 ± 0.16 (WT), 6.44 ± 0.40 (CC2 + hinge), 7.66 ± 0.86 (CC2 + IAK), 9.65 ± 0.85 (hinge + IAK), 10.49 ± 0.57 (CC2 + hinge + IAK). N = 2; n = 25, 15, 12, 20, and 32 MTs. One-way ANOVA followed by Dunnett's test. **p<0.01, ***p<0.001, ****p<0.0001.

The online version of this article includes the following source data and figure supplement(s) for figure 5:

**Source data 1.** XL-MS data for KIF5B (1-565).

**Source data 2.** XL-MS data for KIF5B_IAK_AAA.

**Figure supplement 1.** The landing rate, dwell time, and velocity distribution of all KIF5B variants and the crosslinking mass spectrometry (XL-MS) results of KIF5B(1–565).

**Figure supplement 2.** IAK mutation does not relieve KIF5B folding.

**Figure supplement 3.** Description of KIF5B constructs used in the single-molecule motility assay.

Previous work suggested that mutation or truncation of the tail domain was sufficient for activating kinesin-1 (*Coy et al., 1999*; *Hackney and Stock, 2000*; *Kelliher et al., 2018*; *Seiler et al., 2000*). We thus probed the contribution of the tail domain to autoinhibition by comparing the motility of KIF5B(1–420), our most active KIF5B in terms of landing rate, to that of three KIF5B constructs with altered tail regions: (i) mutation of the IAK segment (IAK construct = QIAKPIR to AIAAAIA), (ii) removal of the entire tail domain (construct 1–909), or (iii) removal of the tail domain plus the C-terminal region of CC4 that crosslinks (K903 and K909) to the motor domain (construct 1–890) (*Figure 5—figure supplement 3*). While all three tail constructs showed an increased landing rate when compared to the full-length KIF5B (*Figure 5B and C*, *Figure 5—figure supplement 1A*), confirming that the tail is critical for autoinhibition, their landing rates did not approach the level of KIF5B(1–420) (*Figure 5B and C*, *Figure 5—figure supplement 1A*), suggesting that removal of the tail is not sufficient to relieve autoinhibition. Interestingly, the removal of the whole tail increased the landing rate more than the IAK mutation (*Figure 5—figure supplement 1A*), suggesting that elements besides IAK in the tail play a role in regulating kinesin-1 activity. When closely examining the dwell time, we found that tail truncation constructs but not the IAK mutant resulted in less dwell time on microtubules compared to full-length KIF5B (*Figure 5—figure supplement 1B*), indicating additional microtubules binding site in the tail, as suggested in earlier studies (*Navone et al., 1992*; *Wong and Rice, 2010*).

As mutation of the tail region was insufficient to relieve autoinhibition, we examined whether disrupting CC2-motor interactions, the hinge between CC3a and CC3b, or the linker between CC3b and CC4 can activate kinesin-1, as suggested by our autoinhibition model (*Figures 2A and 4A*). For CC2, we replaced the sequences that crosslinked to the motor domain (aa 418–452) with a leucine zipper sequence predicted to maintain the coiled-coil nature of this region (construct CC2) (*Figure 5A*, *Figure 5—figure supplement 3*). For the hinge, we replaced the sequences between CC3a and CC3b (aa 678–701) with a leucine zipper sequence predicted to generate an extended coiled-coil for CC3 and CC4 and thereby prevent its folding back (construct hinge; *Figure 5A*, *Figure 5—figure supplement 3*). For the linker, we surmised that increasing the length and flexibility of the linker between CC3b and CC4 (*Figure 2—figure supplement 1*) by inserting the 10 aa sequence GSGGS-GGSGS would reduce motor-CC4 interactions and thus activate kinesin-1 (construct linker)(*Figure 5A*, *Figure 5—figure supplement 3*). Compared to the full-length KIF5B, the CC2 and hinge mutants had increased landing rates but were still dramatically inhibited compared to the fully active KIF5B(1–420) protein (*Figure 5C*, *Figure 5—figure supplement 1A*), suggesting that the release of motor-CC2 and intra-stalk interactions achieves only a partial activation. In contrast, the linker mutant showed similar motility properties as the WT protein, suggesting that lengthening the region between CC3b and CC4 and/or making it more flexible did not alter the folded and autoinhibited state of the molecule (*Figure 5C*).

Since alterations that disrupt only one of the KIF5B intramolecular interactions were insufficient to fully relieve the autoinhibited state, we tested whether combinations of mutations could activate KIF5B motility. We first tested a double mutation of CC2 and hinge, hypothesizing that mutation of motor-CC2 interactions and prevention of CC3 folding would make the motor–tail interactions insufficient for autoinhibition. Although this construct showed a higher landing rate than the full-length KIF5B, it could not achieve the fully activated state of the KIF5B(1–420) protein (*Figure 5D*,

*Figure 5—figure supplement 1A*). Similar results were obtained for double mutations of the CC2 with IAK and hinge with IAK (*Figure 5D*, *Figure 5—figure supplement 1A*). All of the double mutants showed landing rates and dwell times similar to the single IAK mutant (*Figure 5—figure supplement 1A and B*). We thus made a triple mutant that should release all sequences involved in hierarchical folding (CC2, hinge, and IAK) and found that the resulting construct could achieve a landing rate that approaches the tail truncation constructs (1–890 and 1–909), although it is still less active than the fully truncated (1–420) protein (*Figure 5D*, *Figure 5—figure supplement 1A*). These results suggest that multiple interactions contribute to the hierarchical folding of kinesin-1 and that all of these interactions need to be disrupted to fully activate microtubule-based motility.

## The cargo adaptor TRAK1 destabilizes the folded conformation of kinesin-1

Since our modeling and mutagenesis suggest that hierarchical folding is required for kinesin-1 auto-inhibition, we wanted to determine whether an activating adaptor disrupts the folding of kinesin-1. We selected the mitochondrial adaptor protein TRAK1 (*Figure 6A*) as it has been shown to activate kinesin-1 motility (*Canty et al., 2021*; *Fenton et al., 2021*; *Henrichs et al., 2020*). Previous work showed that the N-terminal coiled-coil of TRAK1 is sufficient to bind and activate kinesin-1 in vitro (*Canty et al., 2021*; *van Spronsen et al., 2013*).

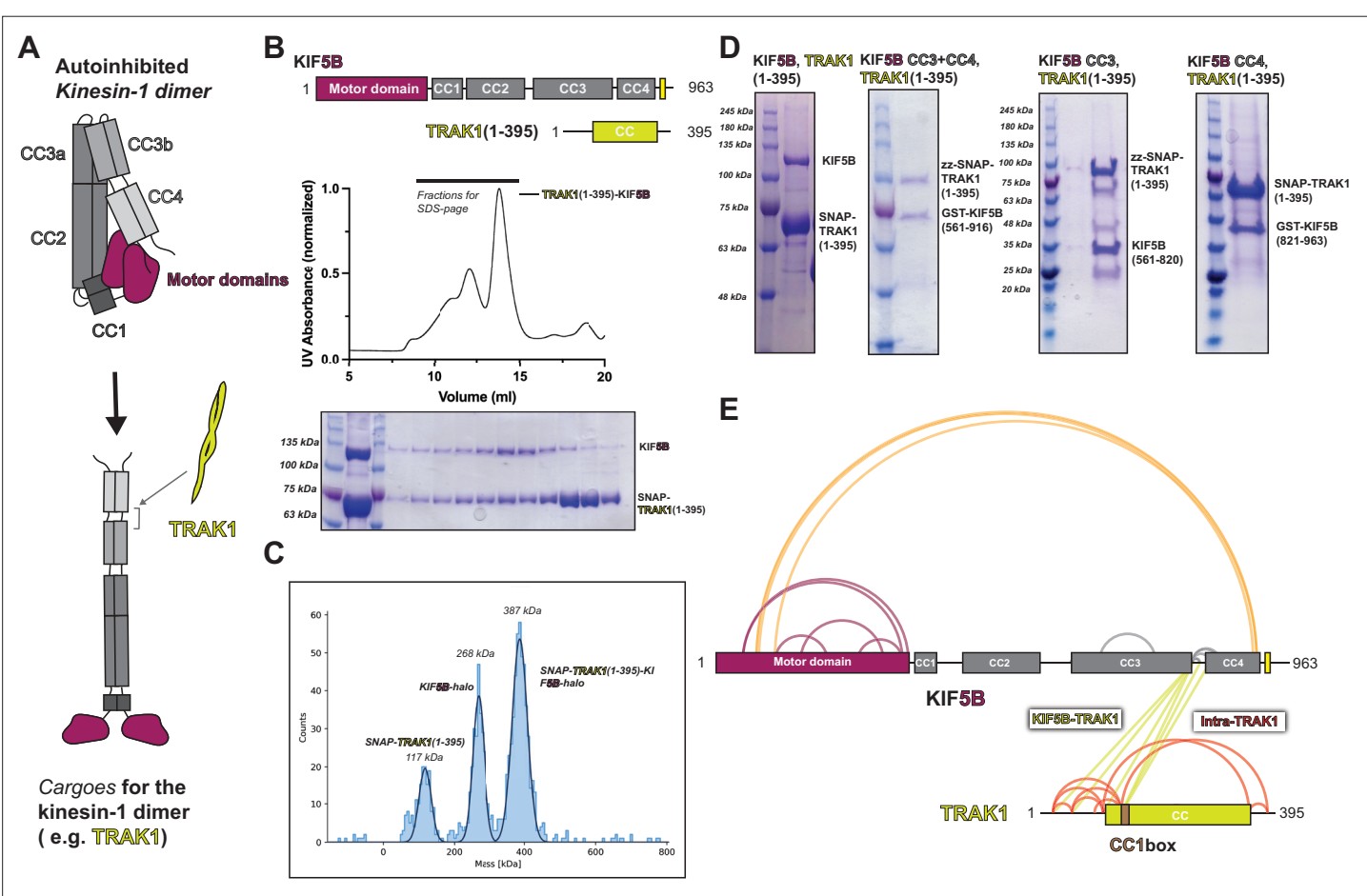

**Figure 6.** TRAK1 interacts with KIF5B with the same location as KLC1 and opens up KIF5B. (**A**) Proposed model of TRAK1 opening up KIF5B. (**B**) The size-exclusion chromatography profile of TRAK1(1–395)-KIF5B complex and the corresponding SDS-PAGE analysis. (**C**) The in vitro pull-down assay between different KIF5B fragments and TRAK1(1–395). (**D**) The mass photometry result for the TRAK1(1–395)-KIF5B complex. (**E**) The crosslinking mass spectrometry (XL-MS) results of TRAK1(1–395)-KIF5B.

The online version of this article includes the following source data for figure 6:

**Source data 1.** XL-MS data for TRAK1-KIF5B.

We first confirmed the complex formation and stoichiometry of the TRAK1-kinesin-1 interaction. Using size-exclusion chromatography, we found that TRAK1(1–395) and KIF5B coeluted as a single species (*Figure 6B*). Then, we measured the stoichiometry of the complex using mass photometry and determined that one TRAK1(1–395) dimer binds to one KIF5B dimer (*Figure 6C*). To further map the binding interface between TRAK1 and kinesin-1, we performed a series of in vitro pull-down experiments using TRAK1(1–395) and KIF5B truncations. The pull-down data showed that TRAK1(1–395) binds to both CC3 and CC4 in KIF5B (*Figure 6D*). Taken together, these data show that a single TRAK1 homodimer binds to KIF5B at the interface of CC3 and CC4.

To probe the conformation of kinesin-1 after binding TRAK1(1–395), we performed XL-MS analysis of the purified TRAK1(1–395)-KIF5B complex (*Figure 6E*). There was a dramatic reduction in crosslinks within KIF5B (compare *Figure 6E* to *Figure 1F*). For KIF5B itself, we noted the disappearance of crosslinks within the stalk (CC2-CC3-CC4) and crosslinks between the motor domain and CC2 (*Figure 6E*). Our XL-MS data also showed that the N-terminus of TRAK1 (K23 to K138) crosslinks to KIF5B between CC3b and CC4 (*Figure 6E*). This binding site is consistent with our pull-down experiments, providing a higher-resolution description of the interaction. Together, our biochemical data and XL-MS suggest that TRAK1 may destabilize the folded, autoinhibited conformation to promote a more open conformation by binding at the CC3b-CC4 region of the KIF5B stalk.

## Discussion

Our integrative modeling approach provides the first comprehensive description of the kinesin-1 autoinhibition mechanism (*Figure 7*). Overall, our results point to hierarchical intramolecular interactions that generate a folded autoinhibited state of kinesin-1, and several important findings can be highlighted.

First, kinesin-1 does not simply fold in half via the originally proposed hinge between the CC2 and CC3a (*Coy et al., 1999*; *Friedman and Vale, 1999*). Our results demonstrate that this previously proposed hinge is instead a highly structured region that adopts a short antiparallel coiled-coil in both the dimeric (KIF5B and KIF5C) and tetrameric (KIF5B/C+KLC1) states (*Figure 2—figure supplement 1* and *Figure 4—figure supplement 1*). This finding explains why deletion of the hinge did not result in an extended conformation for *Drosophila* KHC (*Coy et al., 1999*) and why removal of the hinge resulted in an only a partial increase in kinesin-1 activity (*Chiba et al., 2022*; *Friedman and Vale, 1999*; *Kelliher et al., 2018*). Our results also demonstrate that the folding of kinesin-1 depends on a short (~4 aa) break in CC3, which serves as a hinge that allows CC3b and CC4 to interact with CC2 and CC3a. Our results are consistent with and extend the work of *Weijman et al., 2022*. Surprisingly, mutation of this hinge (called the elbow by *Weijman et al., 2022*) to generate an extended coiled-coil across CC3 did not fully relieve autoinhibition in single-molecule assays. We interpret this result to mean that other regions of KIF5B can generate breaks in the coiled-coil stalk region that enable intramolecular interactions between the motor domain, stalk, and tail.

Second, we identified a novel role for CC2 in the autoinhibition of the two motor domains. In the folded state of the full-length KIF5B model, the two motor domains fold back, and one of them docks on top of CC2 in a manner that blocks the microtubule-binding interface of the motor domain. Positioning of one motor domain along CC2 also enables the C-terminus of CC4 and the tail to be positioned in between the two motor heads and contribute to autoinhibition. In the context of the truncated (1–565) protein, the motor-CC2 interaction results in a protein that, while active in single-molecule motility assays, is still inhibited when compared to the minimally dimeric construct (1–420).

Third, we confirm and extend the role of the tail domain, particularly the IAK motif, in kinesin-1 autoinhibition. Early studies suggested that the IAK motif plays an important role in autoinhibition based on truncations of the KHC from the C-terminus and a tail-bound motor domain structure (*Hackney and Stock, 2000*; *Kaan et al., 2011*; *Stock et al., 1999*). More recent work, however, showed that mutating IAK to AAA did not cause a substantial increase in processive motility but instead increased the landing rate (*Chiba et al., 2022*). In our hands, mutation of the IAK region resulted in only a partial increase of kinesin-1 activity for *Drosophila* KHC (*Kelliher et al., 2018*) and human KIF5B in cell lysate (*Figure 5*). Furthermore, KIF5B with IAK changed to AAA still adopts a folded conformation under EM and XL-MS (*Figure 5—figure supplement 2*), suggesting that the removal of IAK alone is insufficient to open up kinesin-1. Since our purified IAK mutant has mScarlet protein at the C-terminus, we observed an additional globular density in between the motor domains

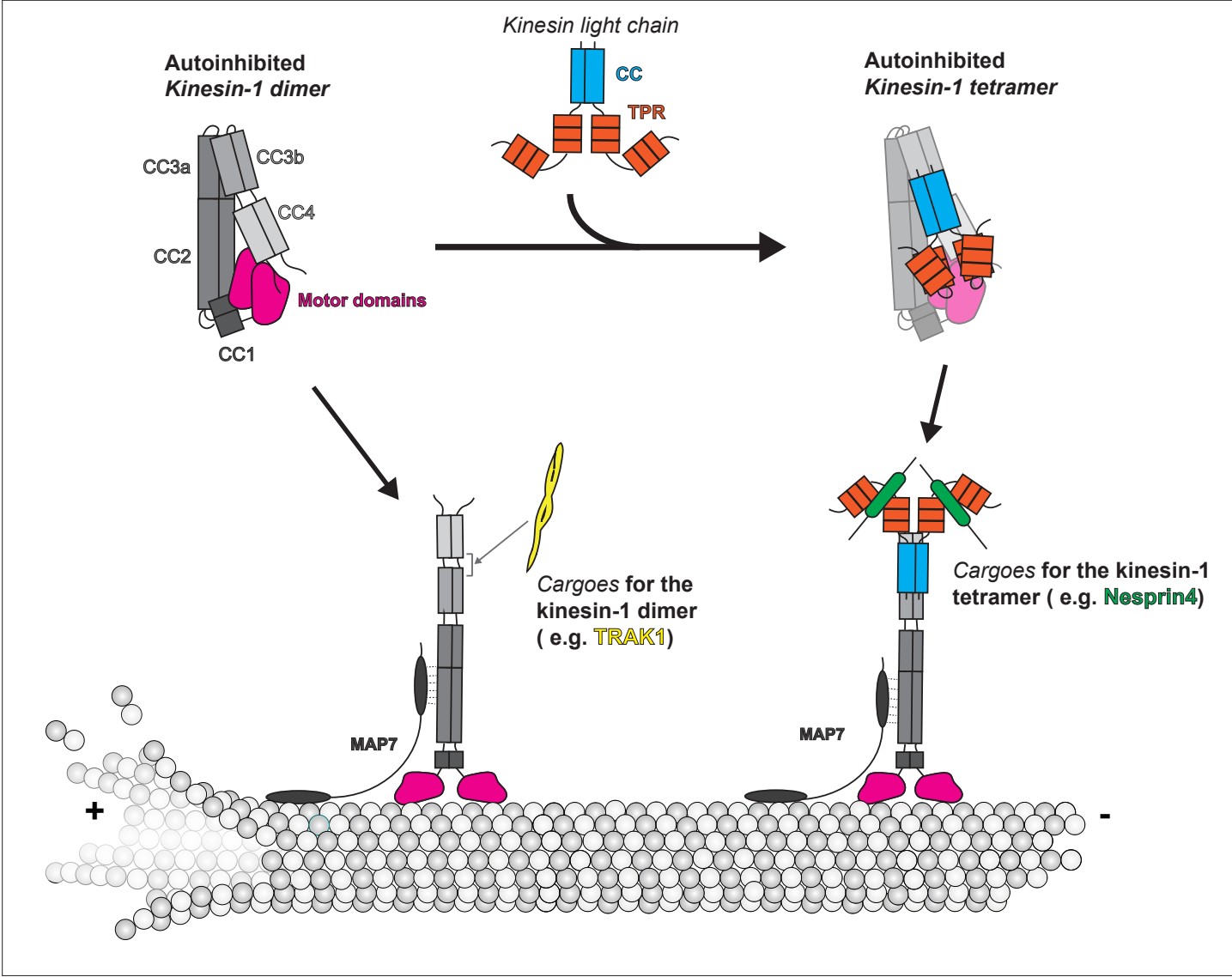

**Figure 7.** Model of kinesin autoinhibition and activation.

(*Figure 5—figure supplement 2B*), indicating that the regulatory tail is still in proximity to the motor domain in the absence of IAK motif.

Interestingly, we observed that deleting the whole tail (constructs 1–890 and 1–909) can achieve a higher landing rate than simply mutating the IAK residues, but the tail deletions showed lower dwell times (*Figure 5B and C*, *Figure 5—figure supplement 1A and B* ) and increased velocities (*Figure 5—figure supplement 1C*). We believe the tail functions as a whole element to regulate kinesin-1 activity. It may stabilize the autoinhibited state of kinesin-1 and reduce the off rate when kinesin-1 walks along microtubules. In our model, the tail domain should lie very close to the motor domain as its upstream CC4 is crosslinked to the motor domains (*Figure 2C*). However, since the tail region of kinesin-1 lacks enough lysine residues and is disordered, we could not observe it via EM and XL-MS, and thus cannot comment on the conformation of the tail and location of IAK motif in the autoinhibited state.

Fourth, the addition of KLC did not change the folding pattern of the heavy chain. From this, we propose that the role of KLCs is to stabilize the folded state of the heavy chain, consistent with previous work (*Chiba et al., 2022*; *Verhey et al., 1998*). Compared to the kinesin-1 homodimer, the addition of the light chain pushes CC4 closer to the motor domain (*Figure 4B*), which may facilitate its insertion in between the motor domain. This finding is consistent with our previous work showing

that the addition of KLC results in a lower FRET efficiency (larger distance) between the two motor domains (*Cai et al., 2007a*) and that the N-terminal coiled-coil region of KLC, not the TPR domain, is required for driving KHC further into the autoinhibited state (*Chiba et al., 2022*; *Verhey et al., 1998*).

Our model of autoinhibition provides a framework to propose the mechanism of kinesin-1 activation by cargoes and MAPs (*Figure 7*). Kinesin-1 binding partners such as JIP3 (*Sun et al., 2011*), TRAK1/2 (*Canty et al., 2021*; *Fenton et al., 2021*; *Henrichs et al., 2020*), JIP1 (*Fu and Holzbaur, 2013*), and HAP1 (*Twelvetrees et al., 2019*) increase kinesin-1 dimer motility in vitro. Here, we selected TRAK1 as our target and further map the TRAK1 binding site on kinesin-1 to be in between the CC3b and CC4 (*Figure 6E*), which overlaps with the KLC binding site, suggesting the incompatibility for simultaneous binding. The association of TRAK1 between CC3b and CC4 may induce a conformational change to release CC4 from motor domains, thus opening the folding within the stalk, as suggested by the decreased number of crosslinks within kinesin-1 after binding to TRAK1 (*Figure 6E*). Besides activating kinesin-1, TRAK1 also serves as a cargo adaptor for the dynein motor (*Canty et al., 2021*). However, the relationship between the kinesin-1 and dynein binding sites is less well understood. Here, after examining the XL-MS data, we realized that the kinesin-1 binding site on TRAK1 is in the upstream region of the CC1 box with only a little overlap with the CC1 box (*Figure 6E*). Given that the dynein–dynactin complex should interact with the downstream of the CC1 box (*Canty et al., 2021*; *Reck-Peterson et al., 2018*), our XL-MS data suggest that there are separate binding sites for kinesin-1 and dynein on TRAK1.

The mechanism of kinesin-1 heterotetramer activation needs further investigation. The TPR domain in KLC functions as a cargo binding platform by binding partner proteins such as Nesprin4 (*Roux et al., 2009*; *Wilson and Holzbaur, 2015*), SKIP (*Pernigo et al., 2013*), and JIP1/3 (*Pernigo et al., 2018*; *Verhey et al., 2001*; *Figure 7*). All these cargoes contain a W-acidic or Y-acidic motif to mediate cargo recognition by KLCs (*Cross and Dodding, 2019*). In cellular assays, a small molecule that mimics the TPR-W-acidic interaction can activate kinesin-1 (*Randall et al., 2017*), and attachment of the W-acidic motif to intracellular cargo can result in kinesin-dependent cargo displacement (*Kawano et al., 2012*). However, in microtubule binding and single-molecule motility assays, binding of the KLC partner proteins JIP1 and SKIP1 was insufficient to activate the kinesin-1 heterotetramer, whereas binding of Nesprin4 could activate when present at saturating concentrations (*Blasius et al., 2007*; *Chiba et al., 2022*). In our model, we placed one TPR domain near the CC2 and the other on top of the motor domain (*Figure 4D and E*), while we also noticed that crosslinked pairs between TPR and KHC are greater than the maximum distance allowed for the BS3 crosslinkers. We propose that cargo binding to TPR may induce a conformational change in KLC that loosens the folding of KHC, the full activation still needs other factors, such as binding a kinesin-1 partner protein (*Blasius et al., 2007*). High-resolution structures of inhibited kinesin-1 heterotetramer are needed for understanding the detailed molecular mechanism of kinesin-1 inhibition and activation.

Full activation of kinesin-1 for cargo transport appears to require not only cargo adaptor proteins but also microtubule-associated factors such as MAP7 (*Figure 7*). In cells, MAP7 and its *Drosophila* homolog ensconsin promote kinesin-1's microtubule association and cargo transport (*Barlan et al., 2013*; *Chaudhary et al., 2019*; *Gallaud et al., 2014*; *Metzger et al., 2012*; *Sung et al., 2008*; *Tymanskyj et al., 2018*). Using in vitro assays, addition of MAP7 directly recruited kinesin-1 to microtubules and can thereby promote the full activation of TRAK1/2-bound kinesin-1 dimer (*Canty et al., 2021*) and Nesprin4-bound kinesin-1 heterotetramer (*Chiba et al., 2022*). Based on our hierarchical folding model for kinesin-1, we suggest that MAP7 facilitates the activation of folded kinesin-1 by removing the inhibitory effects of CC2 on the motor domain. This hypothesis is supported by the findings that (i) MAP7 binds residues 418–526, which span CC2, of KHC (*Hooikaas et al., 2019*), and (ii) MAP7 can increase the motility of KIF5B(1–560) (*Hooikaas et al., 2019*), which remains in a partly inhibited state (*Figure 5—figure supplement 1D*). MAP7's ability to promote kinesin-1 activation is likely also facilitated by its ability to bind to the microtubules through its N-terminal microtubule-binding domain (*Hooikaas et al., 2019*).

In summary, our work combined AlphaFold structure prediction, XL-MS, and EM to obtain a comprehensive description of the autoinhibition mechanism of kinesin-1. We revised the previous model by showing that autoinhibited kinesin-1 adopts a hierarchical folding pattern, where multi-layer intramolecular interactions are needed to promote kinesin-1 fold into the inhibited state. The activation of kinesin-1 thus required multiple protein partners to release the intramolecular folding.

Given the limitations of our studies, we did realize that some crosslinked residues in our model have distances greater than the maximum distance allowed for the BS3 crosslinkers, especially for the crosslinked pairs between TPR and heavy chains, which suggest that our current model could be partially incorrect in specific positional details. Furthermore, we could not fully resolve the role of the tail and TPR domain in maintaining the autoinhibited state. High-resolution structures of kinesin-1 homodimers and heterotetramers are needed to completely understand the autoinhibition and activation mechanisms.

## Materials and methods

### Plasmid construction

For recombinantly expressed kinesin-1 in insect cells, full-length human KIF5B (sp|P33176) and KIF5C (sp|O60282) were synthesized and cloned into the pACEBac1 vector with a C-terminal TEV-ZZ tag. Full-length human KLC1 (sp|Q07866) was synthesized and cloned into the pACEBac2 vector. For the kinesin tetramers, KLC1 was first cloned into the pIDS vector. The pIDS-KLC1 vector was then fused with either pACEBac1-KIF5B or KIF5C via Cre-Lox recombination.

For KIF5B variants used in single-molecule motility assay, all KIF5B mutations and truncations were made through substitution with gene fragments from IDT via HIFI DNA assembly kit (NEB) in a KIF5B mammalian expression vector. All constructs have a linker GGSTLE after the KIF5B sequence followed by mNeonGreen. The KIF5B-IAK/AAA in pACEBac1 vector with a C-terminal mScarlet-StrepII tag used for XL-MS was a gift from RJ McKenney. The KIF5B(1–565) used for XL-MS was made by cloning KIF5B(1–565) into pFastBac1 vector with a C-terminal TEV-ZZ tag.

For KIF5B used in the pull-down assay with TRAK1, the corresponding stalk fragments were synthesized from IDT or amplified from pACEBac1-KIF5B, assembled with GST tag from pGEX6P plasmid and inserted into pACEBac1 vector via HIFI DNA assembly kit (NEB). The full-length KIF5B in pACEBac1 was replaced with a C-terminal TEV-Twin-Strep-tag via HIFI DNA assembly kit (NEB). The TRAK1 construct was made through assembling His-ZZ-TEV-SNAP tag and TRAK1(1–395) into a pACEBac1 vector via HIFI DNA assembly kit (NEB). All constructs were verified using whole plasmid sequencing from Plasmidsaurus.

### Protein expression and purification

All the kinesin-1 constructs were expressed in the Sf9 cells using the Bac-to-Bac baculovirus expression system (Thermo Fisher Scientific). Sf9 cells were maintained as a suspension culture in serum-free medium (Insect-XPRESS, Lonza) at 27°C.

To generate the bacmids, DH10Bac was transformed with kinesin-1 constructs and the bacmids were extracted using a customized protocol. For generating the P0 baculovirus, 150 µl serum-free media (Insect-XPRESS, Lonza) was added to the 24 deep well plate, mixed with 16 µl FuGENE6 (Promega), and the mixer was incubated for 5 min. Then about 6–7 µg bacmid was added to the wells and incubated for 15 min. After that, 850 µl Sf9 cells ($2 \times 10^6$ cells/ml) were transferred to each well of the plate. The plate was then sealed with a gas-permeable film and incubated at 27°C with shaking at 120 rpm for 4 hr. Finally, additional 3 ml serum-free media with 10% heat-inactivated FBS (Thermo Fisher Scientific) was added to each well. The plate was then resealed and incubated at 27°C with shaking at 250 rpm for 5 d. Five days after transfection, nearly all cells showed signs of infection. The culture media were collected, spun at $1000 \times g$ for 5 min, and filtered through a 0.45 µm syringe filter. The P0 baculovirus were stored in the 4°C fridge and kept from light. Next, 50 ml of Sf9 cells ($2 \times 10^6$ cells/ml) was infected with 100 µl of P0 and cultured for 3 d to obtain P1 viral supernatant. The resulting P1 were used for protein expression. For protein expression, 200 ml of Sf9 cells ($2 \times 10^6$ cells/ml) were infected with 4 ml of P1 virus and cultured for 48–65 hr at 27°C. Cells were harvested, rinsed once in 1× PBS buffer, and flash frozen in liquid nitrogen.

All the kinesin-1 constructs were purified using the same protocol. The frozen cell pellet from 200 ml culture was first thawed in a 37°C water bath and then resuspended in 50 ml kinesin lysis buffer (30 mM HEPES-KOH, pH 7.4, 50 mM $KCH_3COO$, 2 mM $MgSO_4$, 1 mM EGTA, 5% glycerol, 0.2 mM Mg-ATP, 0.1% octylglucoside, 0.5 mM DTT, Roche cOmplete Protease Inhibitor Cocktail EDTA free). Cells were lysed in a 40 ml dounce-type tissue grinder (Wheaton) using 20–30 strokes. The lysate was clarified by centrifugation at 40,000 rpm for 35 min at 4°C (Type 45 Ti Rotor, Beckman Coulter). The

resulting supernatant was mixed with 1 ml pre-washed IgG Sepharose 6 FastFlow beads (Cytiva) in a 50 ml conical tube and incubated on a roller at 4°C for 2 hr. After incubation, the kinesin-1-bound IgG beads were transferred to a disposable column and washed with 10 ml kinesin lysis buffer, 10 ml kinesin buffer (30 mM HEPES-KOH, pH 7.4, 150 mM $KCH_3COO$, 2 mM $MgSO_4$, 1 mM EGTA, 5% glycerol, 0.2 mM Mg-ATP, 0.5 mM DTT), 25 ml kinesin high salt buffer (30 mM HEPES-KOH, pH 7.4, 200 mM $KCH_3COO$, 2 mM $MgSO_4$, 1 mM EGTA, 5% glycerol, 0.2 mM Mg-ATP, 0.5 mM DTT), and 10 ml kinesin buffer. The beads were subsequently resuspended in 3 ml kinesin buffer and mixed with 150 ul homemade TEV protease (2 mg/ml). The resulting mixer was incubated on a roller at 16°C for 3 hr. After cleavage, the beads were removed through an Ultrafree-CL centrifugal filter (MilliporeSigma) and the protein of interest was concentrated to 1–2 mg/ml in a 50 kDa molecular weight cut-off concentrator (Amicon Ultracel, Merck-Millipore). The resulting protein solution was subjected to size-exclusion chromatography on a Superose 6 column (Cytiva) equilibrated in the kinesin buffer. Peak fractions were collected and used for the following experiments.

For the pull-down assays with TRAK1(1–395), KIF5B, and KIF5B(821–963), the TRAK1(1–395) was first captured by the IgG Sepharose 6 FastFlow beads (Cytiva). The beads bound to TRAK1(1–395) were incubated with cleared Sf9 cell lysate that overexpressed GST-tagged KIF5B fragments for 2 hr. After incubation, the KIF5B-bound IgG beads were transferred to a disposable column and washed with 40 ml kinesin buffer. The beads were then resuspended in 2 ml kinesin buffer, mixed with 100 µl homemade TEV protease (2 mg/ml), and incubated on a roller at 16°C for 3 hr. After cleavage, the resulting TRAK1-KIF5B complex was subjected to SDS-PAGE.

The pull-down assays with TRAK1(1–395), KIF5B(561–820), and KIF5B(561–916) were performed in a similar way. The GST-tagged KIF5B fragments were first immobilized on the Glutathione Sepharose 4B resin (Cytiva) and then incubated with cleared Sf9 cell lysate that overexpressed TRAK1(1–395). After incubation, the TRAK1-bound resin was transferred to a disposable column and washed with 40 ml kinesin buffer. The TRAK1-KIF5B complex was then eluted via prescission protease (GenScript) and subjected to SDS-PAGE.

## Crosslinking mass spectrometry

For crosslinking kinesin-1, peak fractions from the size-exclusion chromatography were used to prepare the sample for crosslinking mass spectrometry. 1–2 µM kinesin was prepared in 140 µl kinesin buffer where the total protein amount was less than 40 µg. 2 mg lysine-targeting crosslinker bis(sulfo-succinimidyl)suberate (Bfigure supplement 3) (Thermo Fisher Scientific) was dissolved in 500 µl kinesin buffer to a final concentration of 6.9 mM. Three reaction mixtures were prepared with 200×, 400×, and 600× excess of Bfigure supplement 3. The reaction mixtures were incubated at 4°C for 30 min, after which they were quenched with Tris-HCl (pH 7.5) at a final concentration of 25 mM. For each reaction, a 5 µl sample was used for SDS-PAGE to evaluate the crosslinking quality. For crosslinking TRAK1-KIF5B complex, 110 µl TRAK1-KIF5B complex eluted from the beads were incubated with 1.2 mM bis(sulfosuccinimidyl)suberate (Bfigure supplement 3) (Thermo Fisher Scientific) for 30 min at 4°C, which was then quenched with Tris-HCl (pH 7.5) at a final concentration of 25 mM. The resulting crosslinked samples were flash frozen by liquid nitrogen and stored at –80°C.

The samples subjected to mass spectrometry analysis were processed as described earlier (*Solon et al., 2021*). After collecting the raw data, raw mass spectrometry files were searched using pLink 2.3.11 (*Chen et al., 2019*) against target-decoy FASTA files having the sequences of interest and contaminants. pLink was configured for conventional crosslinking with Bfigure supplement 3 linkers. The search was restricted to peptides containing trypsin digestion sites and having up to three missed cleavages. The peptide mass was set from 500 to 6000, and the peptide length set from 5 to 60. The precursor tolerance and fragment tolerance were set to 20 ppm. Search results were uploaded to xiView (*Graham et al., 2019*) for inspection and interpretation.

## Negative stain electron microscopy

Crosslinked kinesin-1 from size-exclusion chromatography was used for negative stain electron microscopy. Kineisn-1 was diluted in the kinesin buffer to 15–20 nM and applied on a glow-discharged carbon grid (Carbon Film 400-mesh copper, Electron Microscopy Sciences). After 1 min incubation, the grid was flipped and sequentially touched to three drops of 0.75% uranyl formate (Electron Microscopy

Sciences) solution for 2 s, 10 s, and 20 s. The staining solution was then left to sit on the grid for 20 s and blotted away with filter paper (Whatman).

Sample screening and data collection were performed with Leginon (*Suloway et al., 2005*) on a Tecnai T12 transmission electron microscope operating at 120 keV equipped with a Gatan Rio camera. Roughly 200–600 images were collected for each kinesin-1 sample in a pixel size of 1.45 Å per pixel. The resulting images were imported into Relion 4.0 (*Kimanius et al., 2021*). CTF estimation was performed using CTFFind4 (*Rohou and Grigorieff, 2015*) in Relion. Particles corresponding to kinesin were picked manually in Relion, resulting in a total of 5000–20,000 particles for each kinesin sample. The particles were extracted in a box size of 580 Å and subjected to the reference-free 2D classification in Relion 4.0 with the new gradient-driven algorithm and a box size of 580 Å. After a few rounds of 2D classification in Relion to clean the particles, the resulting particles were imported to CryoSPARC (*Punjani et al., 2017*) for the ab initio reconstruction. The best map from the ab initio reconstruction in CryoSPARC was imported back as a reference to Relion for 3D refinement to obtain a final map. For the final resolution, the KIF5B homodimer is 29 Å and the KIF5C-KLC1 heterotetramer is 30 Å based on the gold-standard Fourier shell correlation.

Regarding the length measurement, we measured the length of individual molecules that show clear kinesin-1 particles in the micrographs. Molecules that show any sign of aggregation were not measured.

## AlphaFold structure prediction and kinesin-1 modeling

All structure predictions on kinesin-1 were performed using the AlphaFold Multimer (*Evans et al., 2022*) or the ColabFold (*Mirdita et al., 2022*) on the COSMIC² platform (*Cianfrocco et al., 2017*). We tuned the number of recycles to obtain predicted structures. AlphaFold generates several possible models during the protein structure prediction process. These models are ranked based on their confidence scores, which reflect the degree of certainty with which AlphaFold has predicted each model. In our study, we chose the model with the highest score, while we noticed that the top 5 models from the AlphaFold prediction generally tend to be very similar in the case of the kinesin-1 structure prediction.

For the KIF5B dimer model building, we divided the KIF5B into four parts at the end of each coiled-coil domain: (1) motor-CC1-CC2 (1–540), (2) CC2-CC3a (401–690), (3) CC3b (691–820), and (4) CC4-tail (821–963). Each part was predicted separately, and the highest ranked model was selected. We then manually combined the four pieces together based on the distance restraints from the XL-MS and EM map. Given that our XL-MS cannot distinguish which chain in the dimer the crosslinked residue comes from, we assigned the crosslinked pairs to have the shortest distance. The crosslinked pairs were displayed as pseudobonds in the UCSF ChimeraX (*Pettersen et al., 2021*).

For the KIF5C-KLC1 tetramer model building, we first made the model for the KIF5C-KLC1(CC). We predicted the structure of the KIF5C stalk plus the KLC1(CC) and used it as a reference. We then divided the KIF5C-KCL1(CC) into three segments: (1) motor-CC1-CC2 (1–540), (2) CC2-CC3a (401–690), and (3) CC3b-CC4-tail-KLC1(CC), and predicted them independently. These three fragments were then combined together manually based on distance restraints from XL-MS and EM map. We assigned the crosslinked pairs the same way as above and plotted them in the UCSF ChimeraX. For modeling the TPR domains, we used the crystal structure of the KLC1 TPR domain (PDB:3NF1) (*Zhu et al., 2012*) and placed them into the positions with extra densities in the EM map. We assigned the crosslinked pairs to the two TPR domains based on the distance restriction.

## Cell culture, transfection, and lysis

COS-7 (monkey kidney fibroblast) cells obtained from ATCC (RRID: CVCL_0224) were cultured in DMEM (Gibco) with 10% (vol/vol) Fetal Clone III (HyClone) and 1% GlutaMAX (Gibco) at 37°C with 5% $CO_2$. Cells are checked annually for mycoplasma contamination and were authenticated through mass spectrometry (the protein sequences match those in the *Ceropithecus aethiops* genome).

COS-7 cells were transfected with Trans-IT LT1 (Mirus) according to the manufacturer's instructions and collected 48 hr post-transfection. The cells were harvested by low-speed centrifugation (1500 × *g*) for 5 min at 4°C. The pellet was rinsed once in PBS and resuspended in ice-cold lysis buffer (25 mM HEPES/KOH, 115 mM $KCH_3COO$, 5 mM $NaCH_3COO$, 5 mM $MgCl_2$, 0.5 mM EGTA, and 1% Triton X-100, pH 7.4) freshly supplemented with 1 mM phenylmethylsulfonyl fluoride, and protease inhibitors

(P8340; Sigma-Aldrich). After the lysate was clarified by centrifugation at 20,000 × g for 10 min at 4°C, aliquots of the supernatant were snap-frozen in liquid nitrogen and stored at −80°C until further use.

The concentration of KIF5B-mNG proteins in the lysates was measured by dot-blot, in which dilutions of COS-7 lysates expressing KIF5B-mNG WT or mutant proteins and standards of KIF5C(1–560)-mNG protein were spotted onto a nitrocellulose membrane. The membrane was air-dried for 1 hr and immunoblotted with a primary antibody against mNeonGreen tag (Chromotek) at room temperature for 1 hr followed by a secondary antibody (680 nm anti-mouse, Jackson ImmunoResearch Laboratories Inc) at room temperature for 30 min. The fluorescence intensity of the spots on the nitrocellulose membrane was detected by Azure c600 and quantified based on the standard curve of known concentration of KIF5C(1–560)-mNG protein using Fiji/ImageJ (NIH).

### Single-molecule motility assays

HiLyte647-labeled microtubules were polymerized from purified tubulin including 10% Hily647-labeled tubulin (Cytoskeleton) in BRB80 buffer (80 mM Pipes/KOH pH 6.8, 1 mM $MgCl_2$, and 1 mM EGTA) supplemented with 1 mM GTP and 2.5 mM $MgCl_2$ at 37 °C for 30 min. 20 µM taxol in prewarmed BRB80 buffer was added and incubated at 37°C for additional 30 min to stabilize microtubules. Microtubules were stored in the dark at room temperature for further use. A flow cell (~10 µl volume) was assembled by attaching a clean #1.5 coverslip (Fisher Scientific) to a glass slide (Fisher Scientific) with two strips of double-sided tape. Polymerized microtubules were diluted in BRB80 buffer supplemented with 10 µM taxol and then were infused into flow cells and incubated for 5 min at room temperature for nonspecific adsorption to the coverslips. Subsequently, blocking buffer (1 mg/ml casein in BRB80 buffer) was infused and incubated for 5 min. Finally, 1.5 nM kinesin motors in motility mixture (2 mM ATP, 3 mg/ml casein, 10 µM taxol, and oxygen scavenging [1 mM DTT, 1 mM $MgCl_2$, 10 mM glucose, 0.2 mg/ml glucose oxidase, and 0.08 mg/ml catalase] in BRB80 buffer) was added and the flow cell was sealed with molten paraffin wax.

Images were acquired by TIRF microscopy using an inverted microscope Ti-E/B (Nikon) equipped with the perfect focus system (Nikon), a 100× 1.49 NA oil immersion TIRF objective (Nikon), three 20 mW diode lasers (488 nm, 561 nm, and 640 nm) and an electron-multiplying charge-coupled device detector (iXon X3DU897; Andor Technology). Image acquisition was controlled using Nikon Elements software, and all assays were performed at room temperature. Images were acquired continuously every 200 ms for 1 min.

Kymographs were produced from maximum-intensity projections by drawing an ROI along the tracks of motors (width = 3 pixels) using Fiji/ImageJ2. Dwell time was calculated by taking the time of the event along the y-axis of the kymograph. Some motility events that started before image acquisition or finished after image acquisition were included in the analysis, and thus the motility parameters for motors with long dwell times are an underestimate of the true values. Velocity was defined as the distance on the x-axis of the kymograph divided by the time on the y-axis of the kymograph. Full-length motors frequently pause during motility events, and thus the motility events analyzed may include several pauses. The landing rate was defined as (events * $\mu m^{-1}$ * $s^{-1}$ * $\mu M^{-1}$). Only motile events that lasted at least three frames (600 ms) were counted.

For the landing rate measurements, we did more than two repeats for the mutants with no apparent difference compared to the WT. But for mutants with significant differences (1–909, 1–420, etc.), we only did it twice. Here are the information regarding the measurements for the landing rate: WT: 4 repeats, 25 MTs, 29 motile events; 1–909: 2 repeats, 20 MTs, 449 motile events; 1–890: 3 repeats, 23 MTs, 490 motile events; 1–565: 2 repeats, 15 MTs, 936 motile events; 1–420: 2 repeats, 14 MTs, 1424 motile events; CC2: 4 repeats, 25 MTs, 98 motile events; hinge: 4 repeats, 25 MTs, 126 motile events; IAK: 4 repeats, 20 MTs, 126 motile events; linker: 2 repeats, 9 MTs, 7 motile events; CC2 + hinge: 3 repeats, 15 MTs, 99 motile events; CC2 + IAK: 3 repeats, 12 MTs, 81 motile events; hinge + IAK: 4 repeats, 20 MTs, 267 motile events; CC2 + hinge + IAK: 4 repeats, 32 MTs, 503 motile events.

For the dwell time and velocity measurements, We also did more than two repeats for most of the mutants. However, we only analyzed enough events needed to do the statistical comparison.

## Acknowledgements

We thank the members of the Michigan Cytoskeleton Supergroup for helpful discussions and critical feedback. We thank RJ McKenney for providing the KIF5B plasmids with IAK mutation. The research

reported in this publication was supported by the University of Michigan Cryo-EM Facility (U-M Cryo-EM) and Proteomics Resource Facility. U-M Cryo-EM is grateful for support from the U-M Life Sciences Institute and the U-M Biosciences Initiative. This work was supported by R01GM141119 (ZT, MAC), R01GM094231 (VB, AIN, SEH, FVL), R35GM131744 (YY, KJV), and S10OD020011.

## Additional information

### Funding

| Funder | Grant reference number | Author |
|---|---|---|
| National Institutes of Health | R01GM141119 | Zhenyu Tan<br>Michael A Cianfrocco |
| National Institutes of Health | R01GM094231 | Felipe Leprevost<br>Sarah Haynes<br>Venkatesha Basrur<br>Alexey I Nesvizhskii |
| National Institutes of Health | R35GM131744 | Yang Yue<br>Kristen J Verhey |

The funders had no role in study design, data collection and interpretation, or the decision to submit the work for publication.

### Author contributions

Zhenyu Tan, Conceptualization, Formal analysis, Investigation, Methodology, Writing - original draft, Writing - review and editing; Yang Yue, Formal analysis, Methodology, Writing - review and editing; Felipe Leprevost, Sarah Haynes, Software, Formal analysis, Investigation, Methodology, Writing - review and editing; Venkatesha Basrur, Investigation, Methodology, Writing - review and editing; Alexey I Nesvizhskii, Supervision, Funding acquisition, Writing - review and editing; Kristen J Verhey, Conceptualization, Formal analysis, Supervision, Investigation, Methodology, Writing - review and editing; Michael A Cianfrocco, Conceptualization, Data curation, Software, Formal analysis, Supervision, Funding acquisition, Investigation, Visualization, Methodology, Writing - original draft, Project administration, Writing - review and editing

### Author ORCIDs

Zhenyu Tan ⓘ http://orcid.org/0000-0001-6491-5806
Kristen J Verhey ⓘ http://orcid.org/0000-0001-9329-4981
Michael A Cianfrocco ⓘ http://orcid.org/0000-0002-2067-4999

Reviewer #1 (Public Review): https://doi.org/10.7554/eLife.86776.3.sa1
Reviewer #2 (Public Review): https://doi.org/10.7554/eLife.86776.3.sa2
Author Response https://doi.org/10.7554/eLife.86776.3.sa3

## Additional files

### Supplementary files
• MDAR checklist

### Data availability
The raw data from our XL-MS experiments were processed using pLink software. The crosslinked peptide pairs identified were outputted from pLink and organized into tabular formats that delineate the positions of the crosslinks. We have included files as Source Data for our manuscript. We also attached protein sequence information for each XL-MS experiment. Below are the instructions on how to interpret the data. KIF5B: We have performed three XL-MS experiments on KIF5B and chose one of them (KIF5B_xlink_sites_1) as representative data to present in our manuscript (*Figure 1F*, *Figure 2*). The XL-MS results are: KIF5B_pLINK_1, 2, and 3 for identified crosslinked peptide pairs;

KIF5B_xlink_sites_1, 2, and 3 for crosslinked positions in tabular format. KIF5C: We performed one XL-MS analysis on KIF5C and presented it in our manuscript (*Figure 1—figure supplement 1F*). The XL-MS results are: KIF5C_pLINK for identified crosslinked peptide pairs; KIF5C_xlink_sites for crosslinked positions in tabular format.KIF5B-KLC1: We have performed two XL-MS on KIF5B-KLC1 and selected one of them ( KIF5B-KLC1_xlink_sites_1) as representative data to show in our manuscript (*Figure 3—figure supplement 1F* and *Figure 3—figure supplement 2C*). The XL-MS results are: KIF5B-KLC1_pLINK_1 and 2 for identified crosslinked peptide pairs; KIF5B-KLC1_xlink_sites_1 and 2 for crosslinked positions in tabular format. KIF5C-KLC1: We have performed two XL-MS on KIF5C-KLC1 and selected one of them ( KIF5C-KLC1_xlink_sites_1) as representative data to show in our manuscript (*Figure 3F*, *Figure 4*, and *Figure 3—figure supplement 2D*). The XL-MS results are: KIF5C-KLC1_pLINK_1 and 2 for identified crosslinked peptide pairs; KIF5C-KLC1_xlink_sites_1 and 2 for crosslinked positions in tabular format. KIF5B (1-565): We performed one XL-MS on KIF5B (1-565) and presented it in our manuscript (*Figure 5—figure supplement 1D*). The XL-MS results are: KIF5B(1_565)_pLINK for identified crosslinked peptide pairs; KIF5B(1_565)_xlink_sites for crosslinked positions in tabular format.KIF5B_IAK_AAA: We performed one XL-MS on KIF5B (IAK/AAA) and presented it in our manuscript (*Figure 5—figure supplement 2C*). The XL-MS results are: KIF5B_IAK_AAA_pLINK for identified crosslinked peptide pairs; KIF5B_IAK_AAA_xlink_sites for crosslinked positions in tabular format. TRAK1-KIF5B: We conducted one XL-MS analysis on TRAK1-KIF5B and presented it in our manuscript (*Figure 6E*). The XL-MS results are: TRAK1_KIF5B_pLINK for identified crosslinked peptide pairs; TRAK1_KIF5B_xlink_sites for crosslinked positions in tabular format.

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
