## [Editor Report · eLife assessment]

This paper will be of significant interest to the research community working on cytoplasmic transport and microtubule motors, offering **important** insights into the structural arrangement of autoinhibited Kinesin-1. The paper reports a structural model of full-length kinesin-1 describing its autoinhibitory mechanism using cryo-EM, Alphafold structural predictions, cross-linking, and mass spectrometry. The data are of high quality and together offer a **compelling** model for how Kinesin-1 is autoinhibited, indicating that auto-inhibition does not rely on the IAK motif alone but on a more extensive intramolecular interface.

---

## [Referee Report · Reviewer #1 (Public Review)]

Using a combination of structural biology methods, this report aims to describe the auto-inhibited architecture of kinesin 1 either as homodimers or hetero-tetramers. Hence, the multiple contacts between the protein domains and their folding pattern are addressed using cross-linking mass spectrometry (XL-MS), negative stain electron microscopy and Alpha Fold-based structure prediction. Based on the existing literature, the key domains and amino acids responsible for kinesin-1 inhibited state were not clearly deciphered. The synergetic use of different methods now seems to describe in detail the molecular cues that could induce kinesin-1 refolding and opening. Multiple interactions between the different domains seem to induce the folded conformation.

The combination of methodologies is an efficient way to unravel details that could not be addressed previously. The paper is well written. The methods for generating the electron microscopy data and its relevance and quality, for instance, are much better described after revision. In addition, the conclusions are now more convincing because similar investigations are carried out for all isoforms (KIF5B and FIF5C) in parallel.

This article raises the potential strength and power of deep learning structure prediction methods combined simultaneously with other structural biology methods to answer specific questions. In the present context, this study will certainly be helpful in revealing and understanding the activation mechanism of kinesin motor proteins.

---

## [Referee Report · Reviewer #2 (Public Review)]

The authors sought to define the molecular structure of autoinhibited Kinesin-1, which is the major kinesin providing plus-end directed transport on microtubules. The paper reports a structural model of full-length kinesin-1 which builds on the known folded conformation of kinesin-1 and describes its autoinhibitory mechanism using cryo-EM, alphafold structural predictions, cross-linking, and mass spectrometry. The authors study the conformation of dimeric Kinesin Heavy Chain (KHC) and tetrameric KHC bound to the Kinesin Light Chains (KLCs), where KLC stabilizes the autoinhibited conformation. The combination of these various approaches leads to an integrated molecular model of autoinhibited Kinesin-1. Until now, there was some debate over the role of the small coiled-coil 3 (a and b) and where the hinge region of Kinesin-1. The authors resolve this question and present data indicating the hinge is between cc3a and cc3b.

In some places the absence of crosslinks is reported as a lack of interaction, however, it could also be that there are no residues that can be crosslinked in this region. Some crosslinks also are too long to satisfy the model, so it is possible, while most crosslinks occur when Kinesin-1 is inhibited, that a small number of crosslinks arise from when Kinesin-1 adopts another conformation. The structural data are supported by single-molecule motility assays with various mutants of Kinesin-1, which greatly help characterising the domains functionally.

Overall there are some interesting novel data on the autoinhibitory mechanism of Kinesin-1, with well performed and analyzed data with KLC and TRAK. The topic and paper will be of interest to many.

---

## [Author Response]

The following is the authors’ response to the original reviews.

**Reviewer 1**
Major concerns.-The experimental details on the electron microscopy data and more specifically on the processing is too minimal. Because of the missing pieces of information, the data cannot be trusted in its current state. The authors should explain how they processed the data: number of particles, software used, 3D reconstruction algorithms etc...For instance, they do not mention anything about the final resolution and whether they tried to improve it. What is the dimension of the boxes used for 2D classes and 3D reconstruction? Besides, the resulting 3D volumes should be displayed at different orientations or from, at least, a movie so one can see whether the modelled data actually fits into the 3D volume in various orientations. Have the authors tried cryo-EM to improve the resolution of the data? Have they generated 3D classes? Also they should comment on why the resolution if rather low.

Thank you for your valuable feedback on our work. We appreciate your suggestions for improvement and agree that we could provide more detailed information on the experimental details of our electron microscopy data. To address your concerns, we have provided additional information on the processing of the data in the revised manuscript.

Regarding the use of cryo-EM, we attempted to use this technique to determine the structure of autoinhibited kinesin-1. Unfortunately, we encountered challenges in getting the kinesin-1 to behave well on the grids, which prevented us from obtaining meaningful results.

-The report goes back and forth from focusing on KIF5B then KIF5C and back to KIF5B. It is thus confusing for the reader and the rationale for highlighting a specific isoform is not clear. Hence the authors should perform similar analysis for both isoforms. Specifically the alpha fold deed learning modeling should also be performed using KIF5C in parallel with the analysis performed on KIF5B.

Thank you for your feedback on our manuscript. We apologize for any confusion caused by the shifting focus between KIF5B and KIF5C. The KIF5B and KIF5C are both kinesin-1 isoforms, should have high structural similarity and should adopt similar structures.

In our current manuscript, we performed AlphaFold structure prediction on both KIF5B and KIF5C stalks and found that they adopt the same structure. Furthermore, the XL-MS data suggests that KIF5B and KIF5C exhibit similar patterns. We choose to model the KIF5B in this case.

For the kinesin-1 tetramer, we re-performed XL-MS on KIF5B-KLC1 and KIF5C-KLC1 (Author response image 1 and 2) to confirm our analysis in the manuscript. Both data showed that KIF5B-KLC1 and KIF5C-KLC1 have a similar folding pattern. The differences between the two are: (1) The crosslinks within the KIF5B are sparse compared to KIF5C. (2) There are fewer crosslinks between KIF5B and KLC1 compared to KIF5C-KLC1. These differences will need further investigation.Given that there are more crosslinks in KIF5C-KLC1, we choose to model the KIF5C-KLC1 in our manuscript.

**Author response image 1. sa3fig1:** Crosslinked lysine pairs in KIF5B-KLC1 were mapped onto the domain diagram.

**Author response image 2. sa3fig2:** Crosslinked lysine pairs in KIF5C-KLC1 were mapped onto the domain diagram.

-The proportion of compact versus extended form for KIF5B and KIF5C differs. It seems that KIF5B has a higher proportion of compact conformations both as homodimers and heterotetramers? Can the authors comment on this and suggest any possible molecular argument which would induce this difference? Can the authors comment on this discrepancy? What would induce any extended form given that the wild type constructs should be compact only? Is there any equilibrium in solution between the two conformations?

Thank you for your comments on our manuscript. We appreciate your observation that the proportion of compact versus extended form for KIF5B and KIF5C appears to differ. We did observe that KIF5B has a higher proportion of compact conformations both as homodimers and heterotetramers. We have updated our main text and commented on this difference. We do not have a definitive explanation for this difference, but one possibility is that the differences in the sequence of the two isoforms may contribute to their differential propensities for compact versus extended conformations. It is possible that there is an equilibrium between the two conformations, but we did not explicitly investigate this in our study.

In Figure 1.C, lower panel, the "extended" conformation does not appear as extended as stated in the text, looking at the negative stain image. In particular, the one on the bottom right look rather compact, instead. The resulting graph shown in Figure 1.E seems a bit off as compared with the images. How were the measurements performed to generate figure 1.E? Were all the particles selected for measurement or were only some of them picked or were the measurements done using class averages? In the same line, the authors should show class averages of the extended conformation as well.

Thank you for your feedback on our manuscript. We appreciate your comments on the presentation of our data in Figure 1C. We agree that some kinesin may not appear as extended in the negative stain images as we stated in the text. For EM sample preparation, we took the fraction corresponding to the extended conformation, used BS3 to crosslink them and then examined them under EM. The compact kinesin-1 molecule could come from the aggregated molecule during the crosslinking process.

Regarding the measurement, we measured the length of individual molecules which clearly looks like the KIF5B from the raw micrographs. Molecules that show any sign of aggregation were not measured. For the class averages of the extended state, given that the extended molecule is about 80 nm in length and very flexible, it would be hard to get meaningful averages. We have updated the methods section to include this measurement method.

-In figure 2B, the EM envelope does not accommodate the CC1 domain which extends way beyond the contour of the 3D volume and thus suggest that the modeling and/or the 3D EM reconstruction is not correct. Also the authors do not comment at all on this even though this is a striking feature. The CC1 might thereby be less disorganized or more flexible than expected by the model.

Thank you for your feedback on our manuscript, particularly with regard to Figure 2B. We appreciate your observation that the EM envelope does not accommodate the CC1 domain, which extends beyond the contour of the 3D volume. We agree that this is a striking feature that may suggest that the modeling and/or the 3D EM reconstruction is not entirely correct. We have added comments regarding this feature in the main text. However, given the current data, we could not generate a better model to describe the structure of CC1 besides using results from the AlphaFold prediction.

-The so called "C-shaped" feature on the class averages (Fig 3D) does not stand out clearly on all of the class averages. It is visible on the right hand panels but not visible on the left hand side. What is the proportion of classes and thus of the dataset which clearly displayed this peculiar C-shaped feature?? Can the authors analyze this?

Thank you for your feedback on our manuscript, particularly with regard to Figure 3D. We acknowledge your observation that the "C-shaped" feature is not clearly visible on all of the class averages. We believe that it could be due to the different orientations of the class averages. We have revised our main text to comment on this.

-The different mutants were subjected to motility assays. However, mutations/truncations could strongly affect their structural features and conformation. The authors should thus, at least for some of them, check their global ultrastructure using electron microscopy, for instance, and 2D class averaging. In particular, it would be worthwhile testing how different mutations induce any transition from a compact to an extended state. Besides, it is not specified whether the truncated mutants are homo-dimeric or monomeric.

Thank you for your valuable feedback on our manuscript, particularly with regard to the motility assays conducted on the different mutants. All the KIF5B mutants should be homodimers as WT KIF5B. We agree that it would be beneficial to check some of the mutants under EM to examine their conformation. However, due to time constraints, we were unable to perform these analyses.

Minor concernsDoes AlphaFold generate several possible models? Can a selection of those be displayed at least in the supplementary material so the reader can understand how any given model is selected? A short introduction on the alpha fold methodology and how the different obtained structures compare with one another and ultimately how the best structure is selected.

Yes, AlphaFold generates several possible models during the protein structure prediction process. These models are ranked based on their confidence scores, which reflect the degree of certainty with which AlphaFold has predicted each model. In our study, we chose the model with the highest score, while we noticed that the top 5 models from the AlphaFold prediction generally tend to be very similar in the case of the kinesin-1 structure prediction. We have updated the text in the method section to help the reader appreciate our approach.

-When expressing the hetero-tetramers, do the authors generate homodimers as well? If so, can they estimate the relative proportion of all the possible populations?

We used the multibac expression system to co-express the kinesin heavy chain and light chain in sf9 cells. We believe that the hetero-tetramers should account for the majority of products, though we can not rule out the possibility of formation of homodimers.

-The motility assays should be better described.

We have added more text to describe the assay.

-The report does not discuss whether any combinations of isoforms (for instance KIF2B-KIF2C) could assemble into a complex and whether it has already been observed in cells?

We believe that you are asking about whether KIF5B and KIF5C form heterodimer. We did not see any previous literature report on this and have not tested this possibility.

-The authors should discuss why they do not obtain the same results as Kaan et al (2011). For instance, would the experimental conditions responsible for the discrepancies observed?

In the study done by Kaan et al (2011), their structures showed that kinesin-1 motor domains crystallized with a tail peptide holding the motors in an immotile conformation, which supports the model of kinesin-1 autoinhibition where the C-terminal tail of kinesin-1 drives autoinhibition to block motility. However, there are several limitations regarding this study as we mentioned in our manuscript. First, the authors used truncated kinesin heavy chains that only include the motor domain and the neck coil instead of the full length protein. Second, the crystal structure was obtained by adding the tail peptide in trans. Thus, how kinesin-1 folds into an autoinhibited state remains poorly understood, severely limiting our understanding of kinesin-1 regulation.

Our model confirms the critical role of the tail domain as the study done by Kaan et al (2011). We observe that the tail domain lies very close to the motor heads which are consistent with what has been reported in the study done by Kaan et al (2011). However, due to lack of enough lysine residues and the unstructured nature of the tail domain, we could not resolve the exact conformation of the tail domain.

We have addressed the question in our discussion section regarding the tail domain and IAK motif.

-A final schematic model would be beneficial to support the model and could be inserted within the discussion section.

We have added a final model figure as Figure 7 in the discussion section.

-The authors should discuss why the shortest mutant is the most active in the motility assay and how this compares with the full length protein in vivo? Can full-length kinesin1 reach similar motility?

The shortest mutant KIF5B(1-420) only contains the motor domain and CC1, without any regulatory elements to lock it into the inhibited state. It should reflect the intrinsic biophysical property of the kinesin-1 motor domain on the microtubules. We have revised our main text to include this point. However, kinesins in cells are all full length proteins and are subjected to multiple layers of regulation. It would be hard to make the comparison between full length kinesins in vivo and the shortest mutant KIF5B(1-420).

-Have the authors attempted to obtain the structure of a TRAK-1 kinesisn1 complex, for instance by electron microscopy? Will they consider addressing the structure of such full complexes to see whether the protein-protein interactions they infer are indeed reflected within the complexes?

Yes, we did want to check the TRAK1-KIF5B complex using negative staining EM. However, due to the flexibility of TRAK1-KIF5B complex and the low contrast of TRAK1 protein under the negative staining EM, we could not get meaningful results.

-Can the authors test kinesin-TRAK1 complexes in motility assays?

There are already two studies (Canty et al., 2021, Henrichs et al., 2020) that confirmed that TRAK1 can activate the motility of kinesin-1, which we cited in our manuscript. Therefore, we did not test it in our studies.

**Reviewer 2**
-The lack of crosslinks seems to be interpreted as the lack of interactions, but that this is not necessarily the case. Also BS3 crosslinks mainly amino groups that are about 25A apart, which gives a read out of proximity rather than interactions. How many times were the crosslinking experiments done? In figure 6, there are not many crosslinks for TRAK and kinesin-1 so it would be good to know if it has been repeated.

The number of XL-MS we have done for each sample are: KIF5B (three times), KIF5C (once), KIF5B-KLC1 (twice), KIF5C-KLC1 (twice), KIF5B(1-562) (once), KIF5B-TRAK1 (once) and KIF5B(IAK/AAA) (once). We have added the above information in the method section for the XL-MS.

For the kinesin-1 heterotetramers, we re-performed XL-MS on KIF5B-KLC1 and KIF5C-KLC1 (Figure 1 and Figure 2) to validate our analysis in the manuscript, which shows consistent results as in our manuscript. For the XL-MS experiment on the KIF5B-TRAK1 complex, due to the time limitation, we only performed it once but would like to explore it in the future.

We summarized identified cross-linked pairs for each kinesin-1 sample as supplementary files.

-Regarding the interaction between TRAP and Kif5b, the authors propose TRAP activate Kif5b by disrupted the autoinhibited conformation from the lack of crosslinks and the position of the cross-links identified. What does Kif5b+TRAP (after or before crosslinking) look like by negative stain EM? The authors have done this experiments for the other samples Kif5b and Kif5b KLC so it would should be easy for the authors to do this for Ki5f5b-TRAP. Also can alphafold mutimer predict the Ki5fb-TRAP interface?

Thanks for bringing this up. We tried to get the EM images for the TRAK1-KIF5B complex. We observed that the KIF5B alone and the TRAK1-KIF5B complex tend to fall apart if not being crosslinked before putting onto the grids. For the crosslinked samples, we are unable to see the TRAK1 clearly on the KIF5B due to the flexibility of the TRAK1-KIF5B complex and the low contrast of TRAK1 protein under the negative staining EM. We would like to explore this further.

As for the AlphaFold prediction on KIF5B-TRAK1 complex, we found that AlphaFold did not perform well in predicting the TRAK1 on kinesin-1 stalk. We tried the combination of various TRAK1 and KIF5B fragments, but could not get any meaningful results.

-Figure 4. Very long crosslinks are not explained by the model, and suggest the model could be partially incorrect. Can the authors state the distance between the crosslinked residues in their model in figures? Generally the authors should report all crosslink distance in their figures with molecular models.

Thanks for bringing this up. For the model building, we used the XL-MS data as guidance to model the autoinhibited kinesin-1 with the input from AlphaFold structure prediction and EM map. We assembled the model by piecing together multiple rigid kinesin-1 fragments generated from AlphaFold structure prediction as described in the method section.

We realize that some crosslinked residues in our model have distances greater than the maximum distance allowed for the BS3 crosslinkers, especially for the crosslinked pairs between the TPR and motor domain. We admit that our current model could be partially incorrect. Since we do not have high resolution structure data on kinesin-1, we are unsure about how to make our model to satisfy all the distance constraints. We have addressed the above limitations in our discussion section.

-Figure 5: motility assays, the amount of data analyzed seems quite low. There are only 2 repeats done for each condition. The number of microtubules is reported rather than number of measurements done-can the authors report number of events/motors measured. It would be useful to have the concentration of motors used in the figure.Landing rate: are authors not differentiating motile vs non motile tracks also?What do the mutants look like in EM class averages?

Thanks for bringing this up. We have revised our method section about the single molecule assay to include this information.

Finally, we agree that it would be beneficial to check the mutants under EM. However, due to time limitations, we were unable to perform this experiment.

-The figure in 6D needs revising. This does not look like a pulldown experiment, controls are missing and the proteins do not seem to be stoichiometric. In particular, the third lane. There are also no protein markers.

Thank you for bringing this up. We revised Figure 6 and added the protocol for the pulldown assay in our method section for protein expression and purification.

Minor points-Is the data available in PRIDE, etc...? Could the authors provide a table of xlinks?

We have included crosslinked pairs detected in our XL-MS as supplementary files for KIF5B, KIF5C, KIF5B-KLC1, KIF5C-KLC1, KIF5B(1-565), KIF5B(IAK/AAA) and KIF5B-TRAK1. We have added a new section called Data Availability in the main manuscript to fully describe this.

-It would be better to have the mapping of the crosslinks in the same figures as the corresponding crosslink map.

Due to the layout of the figure, we choose to show the model and the mapped crosslinks in the same figure.

-No crosslinks were obtained between the IAK motif and the motor domain. This could be due to the lack of neighbouring groups that can crosslink with the K in the motif, rather than the tail not binding/crosslinking to the motor. The text could be edited to explain this

Thanks for bringing this up. We edited the text to add this point.

-Figure 5. Typo in mutation

We revised the figure5

-No hyphen between c and terminus (as that is a noun)